# Chemical reservoir computation in a self-organizing reaction network

Mathieu G. Baltussen[1], Thijs J. de Jong[1], Quentin Duez[1], William E. Robinson[1] & Wilhelm T. S. Huck[1✉]

Chemical reaction networks, such as those found in metabolism and signalling pathways, enable cells to process information from their environment[1,2]. Current approaches to molecular information processing and computation typically pursue digital computation models and require extensive molecular-level engineering[3]. Despite considerable advances, these approaches have not reached the level of information processing capabilities seen in living systems. Here we report on the discovery and implementation of a chemical reservoir computer based on the formose reaction[4]. We demonstrate how this complex, self-organizing chemical reaction network can perform several nonlinear classification tasks in parallel, predict the dynamics of other complex systems and achieve time-series forecasting. This in chemico information processing system provides proof of principle for the emergent computational capabilities of complex chemical reaction networks, paving the way for a new class of biomimetic information processing systems.

Complex chemical reaction networks are involved in all key processes of life. Of these processes, information processing sits at an important nexus between a cell and its surrounding environment. Signalling pathways process environmental information to coordinate cellular responses, while metabolic networks work to maintain homeostasis in response to the ever-changing surroundings[1,2]. Since the dawn of computer science, researchers have speculated about harnessing the inherent potential of physical and biological systems for computation, a function that can also be formulated as information processing[5–7]. Substantial progress has been made in constructing chemical systems that use Boolean logic[8,9], digital computation[10–13], neural networks[14–18], pattern recognition augmented by in silico deep learning[19] and sequence recognition[20]. Self-learning chemical systems have been theorized for abstract chemical reactions[21,22]. These approaches demonstrate how molecular systems may perform computation, but do not achieve the information processing capabilities of living systems. Unlocking the full potential of molecular systems requires (1) moving beyond a strict adherence to reproducing digital computation principles and (2) finding an approach that overcomes the laborious nature of bottom-up 'molecule-by-molecule' design patterns.

Our recent work on dynamic self-organization of chemical reaction pathways in the formose reaction[4,23] inspired us to consider its propensity for information processing. This complex reaction network produces a rich diversity of possible chemical compositions that are nonlinearly dependent on a small number of input reactants and catalysts. Under flow conditions, the distribution of these compounds can be modulated using changes in reactor input concentrations, allowing a range of complex self-organized reaction responses to be controlled with a relatively simple set of input parameters. These properties, and the experimental tractability of the formose reaction, make it an excellent candidate system for exploring chemical information processing using the model of physical reservoir computation. Physical reservoir computing is part of a family of so-called neuromorphic approaches, which use the analogue and dynamic nature of physical systems to process information and perform computations[24–27]. A range of computational tasks, such as classification[28] and simulation[29,30], have been demonstrated in a variety of materials, such as photonic devices[31], spintronic oscillators[32] and nanowire networks[33].

Here we report on the experimental realization of in chemico computing by establishing that the formose reaction has emergent computing properties, obviating the need for complex bottom-up design and creating new opportunities for scalable molecular computing. We demonstrate that the formose reaction is capable of performing several parallel, nonlinear classification tasks, how it can model the behaviour of complex dynamical systems and how it can perform time series forecasting. Our work shows how chemical reaction networks process information on the basis of self-organization, and, much like biological systems, can achieve a variety of powerful computational tasks using information from their environment.

## A chemical reservoir computer

Our chemical reservoir computer is built around the formose reaction (Fig. 1a) in a continuous stirred tank reactor (CSTR) (Extended Data Fig. 1 and Methods). Following the reservoir computation model[34], we can approximate any target (dynamic) transformation ($f$) under the influence of a set of input variables ($u$) (Fig. 1b,e) by feeding the input variables as a sequence of chemical concentrations into the reservoir (Fig. 1c). We investigated three kinds of target transformations: analytical expressions in the form of classification tasks, integral solutions of differential equations and chaotic maps in the form of time series forecasts of the Lorenz system. The first type of task uses only the steady-state features of the reservoir to approximate a static function, whereas the other types use the full dynamics of the reservoir

[1]Institute for Molecules and Materials, Radboud University, Nijmegen, The Netherlands. ✉e-mail: wilhelm.huck@ru.nl

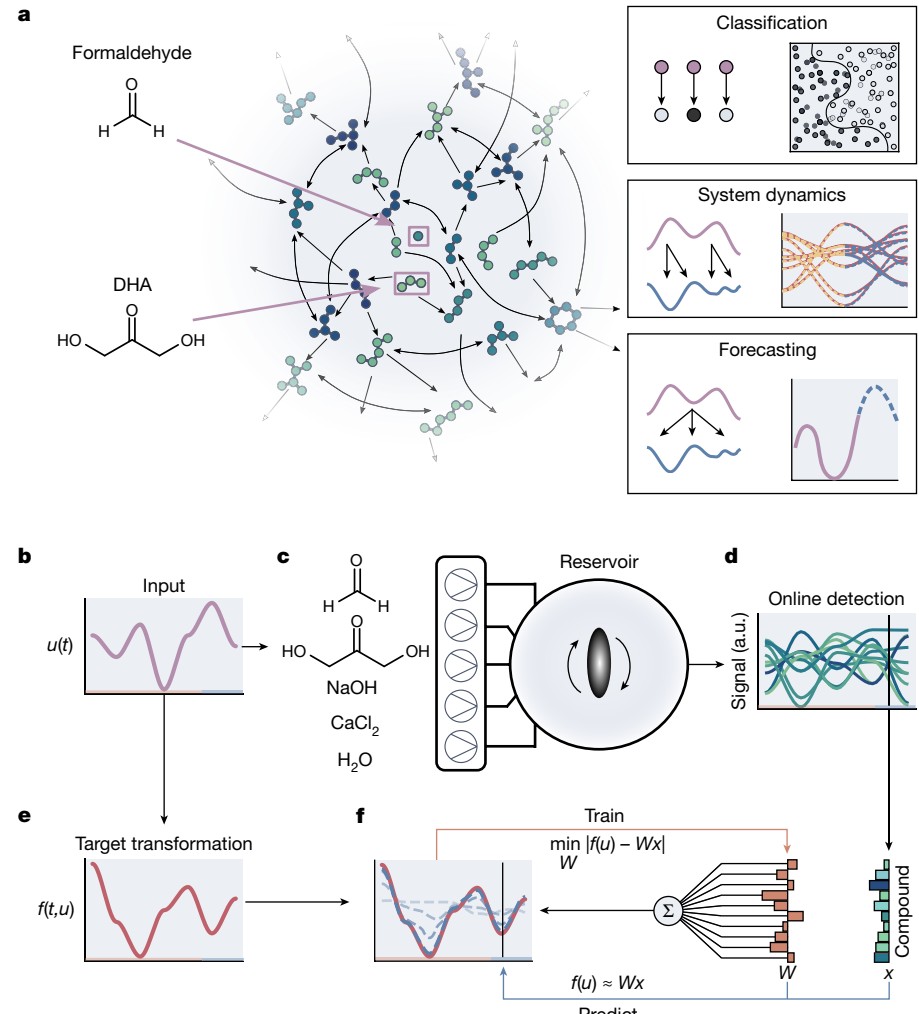

**Fig. 1 | A schematic overview of the formose reservoir computer. a**, The formose reaction and its information processing abilities. Left, a schematic view of the formose reaction network. Arrows indicate chemical transformations between compounds in the network. Dihydroxyacetone and formaldehyde are used as initial reactants and indicated with purple arrows. Right, graphical summary of the information processing tasks of which the formose reaction is capable. **b**–**f**, A schematic overview of the experimental set-up and reservoir training process. A set of input variables *u* used to obtain a target (dynamic) transformation *f*(*t*, *u*) (**b**). These input variables are also used as flow inputs into the reservoir. Syringe pumps containing the formose reagents

(formaldehyde, dihydroxyacetone, sodium hydroxide and calcium chloride) are connected to the inlets of a CSTR and are used to feed the input into the reservoir (**c**). The reservoir outlet is connected to an ion mobility mass spectrometer for online detection of the reservoir composition (**d**). The state of the reactor *x* is measured over time in response to changing inputs. The target (dynamic) transformation *f*(*t*, *u*) obtained from the input (**e**). Weights *W* are trained on the states of the reservoir to obtain an approximation to the target function, which can then be used for further predictions (**f**). a.u., arbitrary units.

to approximate different kinds of dynamic systems. Input concentrations to the reservoir are controlled by changing the flow rates of formaldehyde, dihydroxyacetone (DHA), sodium hydroxide (NaOH) and calcium chloride (CaCl$_2$), making a total of four possible reactor inputs in our set-up. Reservoir outputs are measured by an ion mobility mass spectrometer, from which we extract the relative abundance of up to 106 different ions, characterized by unique mass-to-charge (*m*/*z*) ratios and inverse mobilities, with a time resolution of 500 ms (Fig. 1d, Methods and Supplementary Information section 1). The nonlinear response of the chemical reservoir computer to the input *u*, a collection of ion species denoted by *x*, is recorded and converted to a 'computational' output by training a single linear read-out layer, which multiplies every ion signal with a weight (denoted by *W*) and sums the resulting weighted signals (Fig. 1f). These weights are trained to replicate the target function using a simple linear regression algorithm specific to the computation task (background explanation in Supplementary Information section 2). This single-layer training step is an essential feature of

reservoir computation; it allows us to translate the reservoir response into the desired computation result. By using different sets of weights, the same experimental data can be used to solve several computation tasks. Depending on the computation task, the inputs, reservoir states and outputs may be time dependent or remain constant.

## Nonlinear classification

We first demonstrate how the formose reservoir can chemically process information from its environment (for instance, the concentrations of species flowing into the reactor) and produce a well-defined classification response (for example, 0 or 1). We created a two-dimensional input space consisting of 132 randomly sampled concentrations of formaldehyde and NaOH (10–150 mM and 10–50 mM, respectively), normalized between 0 and 1 (Fig. 2a) while keeping the DHA and CaCl$_2$ inputs constant. This parameter space was chosen based on previous work[4] that demonstrates a complex, hierarchical compositional

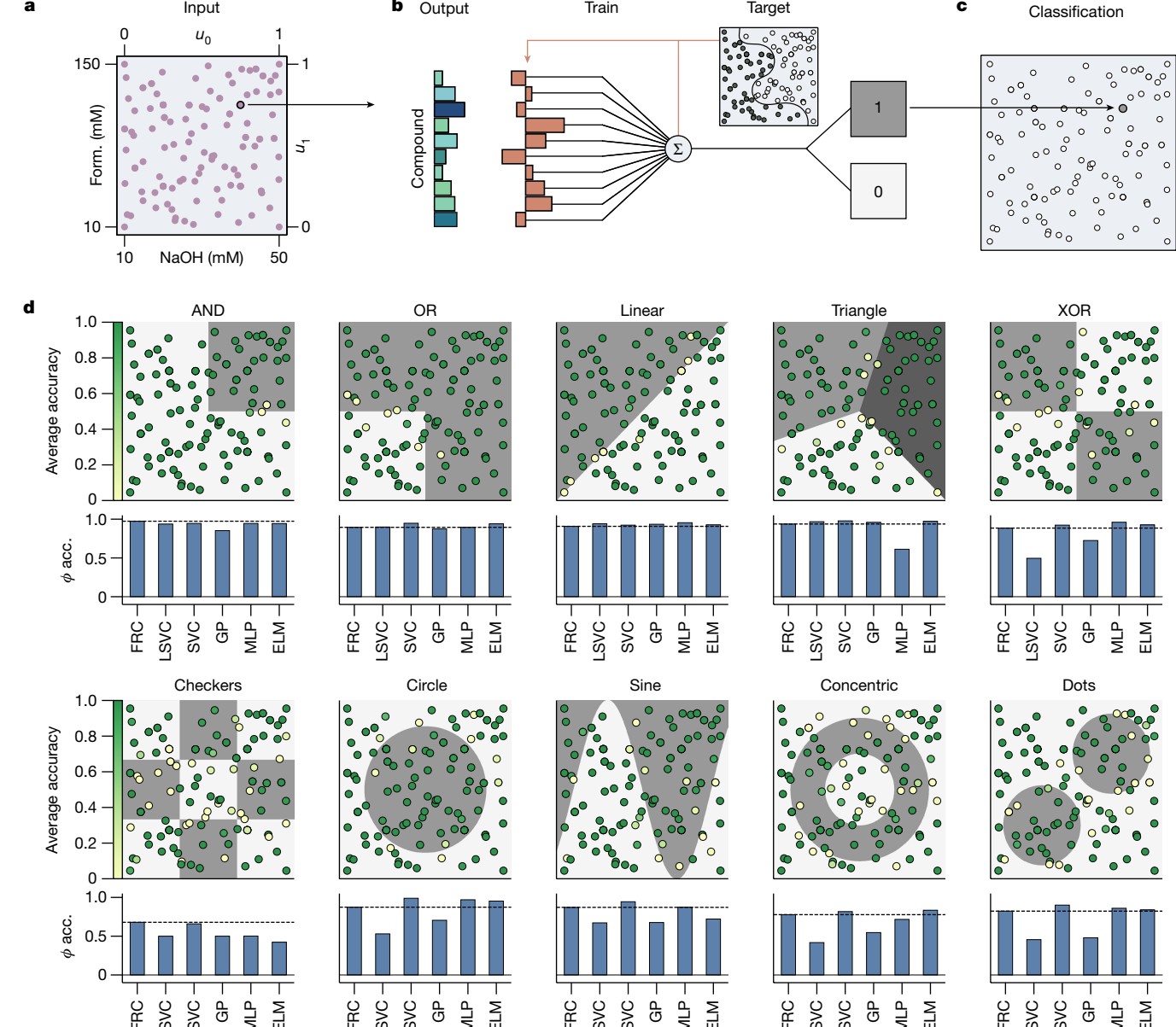

**Fig. 2 | Nonlinear classification. a–c**, Schematic showing the classification of an input combination. **a**, A scatter plot showing combinations of formaldehyde and NaOH concentrations used to create an input space for nonlinear classification problems. **b**, A bar chart representing the steady-state averaged reservoir response (not all compounds shown), associated with selected formaldehyde–NaOH input combinations. Weights are trained on a target classification and applied to the reservoir response. **c**, The response is classified as either a 0 or a 1 on the basis of trained weights. **d**, Results of reservoir classifications for various classification tasks. Dot locations indicate the corresponding input from **a**, with the colour of every point indicating the test-set accuracy of that point for 520 different leave-five-out train-test splits (20 repeats of 26 splits), where +1 corresponds to perfect predictions, and 0 to total failure (see Methods). Shaded areas indicate the different classes of the classification function. The bar chart below every classification plot shows a comparison between the average test-set $\Phi$ accuracy for the formose reservoir (FRC), the training layer without reservoir (LSVC) and various other machine learning classifiers. The dashed line indicates the score achieved by the formose reservoir. acc., accuracy; Form., formaldehyde; GP, Gaussian process classifier.

landscape in this concentration range. For each unique input point in this space, the reactor was allowed to reach a steady-state composition over a 30-minute equilibration period. The final output of the formose reservoir was obtained as the averaged ion intensities over the final 10-minute sample period of the steady-state output, or 1,200 data points per unique input for each of the 106 ion signals (Methods, Supplementary Information section 3.1 and Supplementary Figs. 5 and 6). These data establish a large dataset that is deterministic (Supplementary Information section 3.2 and Supplementary Fig. 7) and robust against outliers and overfitting. We next trained weights on

the 106-dimensional output for every input combination using a linear support vector classifier (LSVC) algorithm (Fig. 2d), resulting in a classification for every unique input (Fig. 2c). This training procedure was performed for a variety of nonlinear classification tasks (Fig. 2d) and validated by calculating the average $\Phi$ accuracy (also known as the phi coefficient, or Matthews correlation coefficient) for 520 different leave-five-out train-test splits (Methods). The reported $\Phi$ accuracies are the averages over all test sets and can be found in Extended Data Table 1.

Notably, the reservoir can emulate all Boolean logic gates and various, more advanced, nonlinear classification tasks, such as sine and

(concentric) circle classifications (Fig. 2d). Such tasks have previously been possible only in molecular systems specifically designed to perform the function of a single logic gate[35]. The formose reservoir is capable of performing any of these nonlinear classification tasks without requiring a redesign of the chemical network. It can therefore perform a broader range of computational tasks than previous molecular systems, enabling considerable computational generalizability. Comparing the formose reservoir to standard nonlinear classification algorithms allows us to further demonstrate the flexibility in information processing available with the chemical reservoir, compared to standard in silico classification methods. For the linear tasks (AND, OR, linear and triangle tasks), the reservoir performs similarly to the in silico algorithms. For the nonlinear tasks (XOR, checkers, circle, sine, concentric circles and dots), the formose reservoir outperforms Gaussian process classification. It scores comparably to support vector classifiers (SVCs), multilayer perceptrons (MLPs) and extreme learning machines (ELMs) for the XOR, checkers, sine, concentric circles and dots tasks, and is only outscored significantly for the circle classification task. Notably, SVCs and MLPs are specifically designed algorithms for such nonlinear classification tasks as shown here, and are therefore expected to perform well. Crucially, the formose reservoir produces similar results to these specialized classifiers solely by tuning a single linear regression head.

## Predicting complex dynamical systems

Living systems can detect, exploit and predict changes in their environment over time. Encouraged by the formose reservoir's flexibility for performing classification tasks in time-invariant settings, we modified our approach towards using it to predict the dynamic behaviour of complex systems in a fluctuating environment. We investigated the ability of the formose reservoir to predict ordinary differential equations typically encountered in fields such as ecology, systems biology, chemistry and engineering. These types of dynamical system are challenging to simulate and predict, especially when mathematical functions are not available to describe them. These systems are often exposed to random fluctuations from their environments, affecting their behaviour in complex manners.

We showcase the formose reservoir's ability to simulate a dynamical model system by fitting it to a carbon-metabolism model of *Escherichia coli*, a large, partially recursive, nonlinear metabolic network with 87 substrates and 92 reactions, adapted from refs. 36 and 37 to include extra inflow and outflow terms. An overview of the training and prediction procedure is shown in Fig. 3a, and a schematic of this network is shown in Fig. 3b (full details in Methods). We perturbed the system using a fluctuating DHA input ($u(t)$) and solved the differential equations of the system to obtain the response in substrate concentrations ($y(t)$). Likewise, we exposed the formose reaction to $u(t)$ (by means of DHA inflow; Extended Data Fig. 2) and used a training period of 30 min (approximately 3,600 data points per ion signal for 106 ion signals) to find the linear mapping that allows the formose reaction to reconstruct the behaviour of the metabolic network (Fig. 3c). Performing the mapping in this manner is equivalent to learning which linear combinations of compounds produced in the formose system best recreate the behaviour of the dynamical system under investigation. By continuing the fluctuating input pattern after the training period, the learned linear mapping allowed us to use the formose reservoir as an emulator of the dynamic system.

In Fig. 3c, time trace comparisons between true and predicted concentrations are shown for pyruvate, 3-phosphoglyceric acid and the co-factor adenosine monophosphate (AMP), showing how the formose reservoir can closely predict the behaviour of the network, for a training time of 30 min and a prediction time of 60 min. Extended prediction times of up to 90 min are shown in Extended Data Fig. 3 and are shown per substrate in Supplementary Figs. 17 and 18. In the metabolic

simulation, the effects of environmental fluctuations vary across the network: they can be linear for substrates close to the environmental inputs, or highly nonlinear for more downstream substrates. The formose reservoir can capture both types of behaviour, using its dynamic properties to correctly incorporate nonlinear and delayed responses. Comparison plots between reservoir prediction and the true in silico behaviour for all substrates are shown in Fig. 3d for different concentration regimes (each colour represents a substrate), showing that most predictions closely match the true behaviour across several orders of magnitude. It is not a perfect predictor, as it inaccurately captures the behaviour of some species that accumulate or break down over very long timescales without being influenced by environmental fluctuations, primarily substrates involved in the glyoxylate (GLX) and acetyl coenzyme A (ACCOA) cycles (Supplementary Figs. 17 and 18). However, this inaccuracy is anticipated, as the formose reservoir does not contain such long reaction timescales. Nevertheless, these results show that the internal nonlinear dynamics of the formose reaction network make it a promising reservoir system for computing the dynamic behaviour of complex (bio)chemical systems.

## Forecasting and mutual information

Autonomous systems, such as bacteria[38], the human brain[39] and self-learning artificial intelligence (AI)[40], can anticipate changes in their environment based on learned experiences to navigate, maintain stability and make decisions. Inspired by the prospect of performing such operations using chemical information processing, we wanted to explore how the formose reservoir's short-term information storage capabilities, or memory, can be harnessed to forecast future environmental dynamics. For environmental dynamics with a temporal structure (for example, deterministic dynamics and/or periodicity), we can attempt to forecast changes by learning a linear mapping between the reservoir state $x(t)$ and a future input as $u(t + \delta t) = Wx(t)$ (shown in Fig. 4a). Here, $W$ denotes the linear mapping in the form of static weights learned during a short training phase, and $u(t + \delta t)$ represents the environmental dynamics a time $\delta t$ into the future. We demonstrate this mapping using a chaotic three-dimensional input, based on the Lorenz attractor, using orthogonal projections of its trajectory to generate three time-dependent inputs into the reactor (DHA, NaOH and formaldehyde), for which we recorded the reservoir response to forecast the inputs 120 s into the future (Methods and Extended Data Fig. 4). A 20 min training period was used; the forecasts for each of the fluctuating inputs are shown in Fig. 4b. For two of the input dimensions (DHA and NaOH), the formose reservoir can accurately and reliably forecast their dynamics for several hours (Supplementary Fig. 20), even while the reactor contents are continuously refreshed. The formaldehyde forecast is less accurate, although the overall dynamics are still correctly predicted. This is probably because the time-dependent response of the formose reaction to increasing formaldehyde concentration is slower than the input dynamics.

These results show that the formose reservoir assimilates information from its input over time, which can be used to leverage correlations in environmental dynamics to predict future environmental changes. To further evaluate the memory properties of the reservoir, we calculated the mutual information, sometimes referred to as predictive information[41], between every compound in the formose reservoir $x(t)$ and the three inputs at varying time delays $u(t + \delta t)$, quantifying the propagation of environmental information through the formose network over time. The relationship between input, ion signals and mutual information is shown schematically in Fig. 4c. We use $\delta t$ as a time-lag parameter, allowing us to calculate the mutual information between reservoir state and past or future inputs according to equation (1) (Methods). The calculated mutual information is shown for a range of lag times into both the past and the future in Fig. 4d for all ion signals, with several compounds highlighted (mutual information per

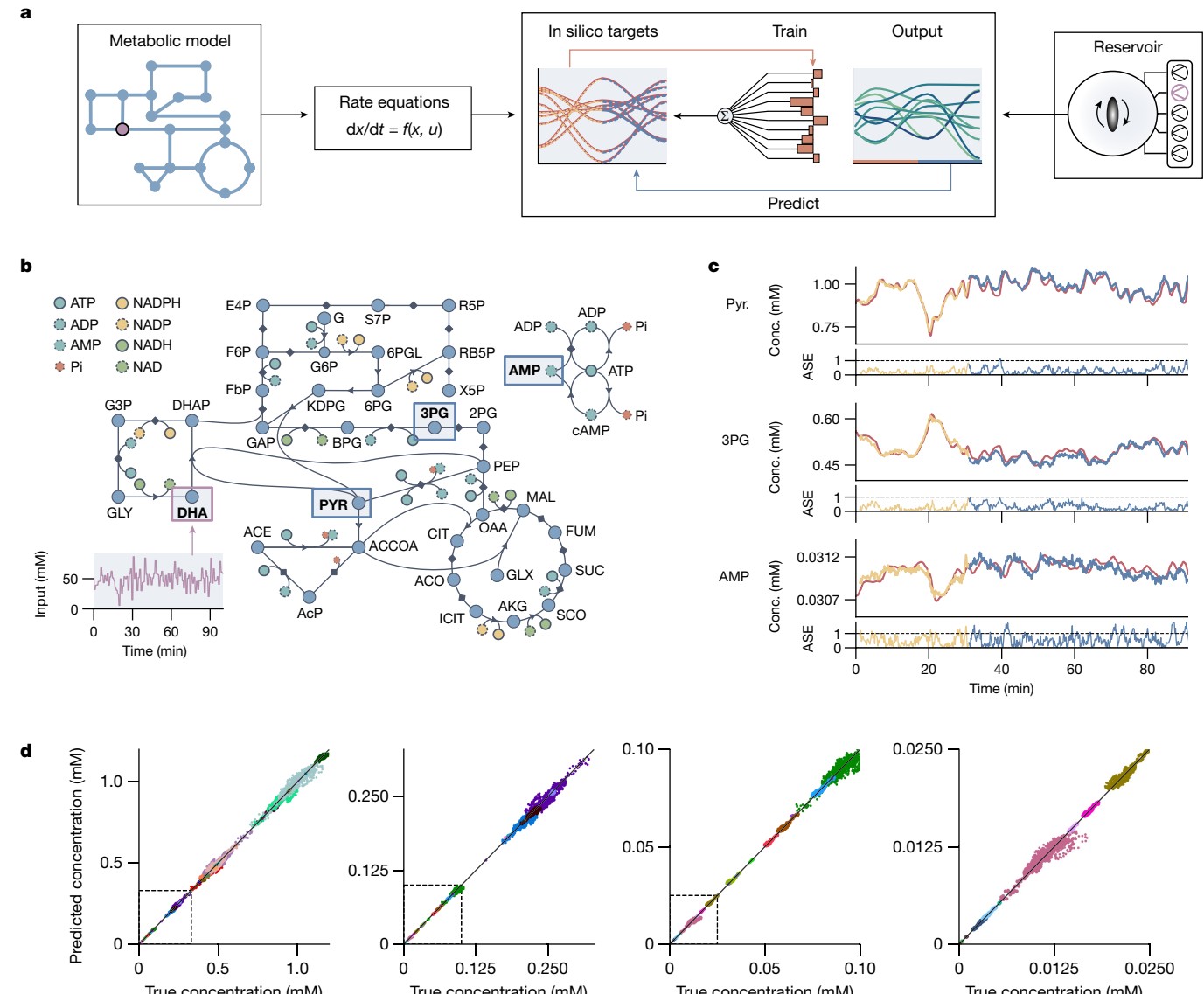

**Fig. 3 | Modelling of complex dynamics. a**, Schematic showing the modelling of a metabolic network. Left, metabolic model with a time-dependent input (purple) is converted to a set of rate equations, which are used to provide an in silico target function. Right, the chemical reservoir uses the same time-dependent input (purple) to produce a complex output over time. Middle, weights are trained to map this output onto the target, which are then used for further predictions. **b**, A schematic representation of the carbon metabolism found in *E. coli*. The fluctuating DHA input is indicated in purple, and the location of the three substrates shown in **c** are indicated with blue rectangles. Arrows indicate irreversible reactions; diamonds indicate reversible reactions. **c**, Three example predictions for the behaviour of pyruvate (Pyr.),

3-phosphoglyceric acid (3PG) and AMP. The red lines indicate the 'true' simulated response of the model. The yellow lines show part of the train set, and the blue lines show the predictions of the formose reservoir after training. Indicated times on the *x* axes are both the physical reservoir time and the model time. The absolute scaled error (ASE) (see Methods) over time is shown below the predictions. **d**, Comparison plots between true and predicted concentrations of various substrates in the carbon metabolism under a fluctuating DHA input. Plots with decreasing concentration ranges are shown left to right, with the dashed region indicating the concentration range of the next plot. Different colours indicate different substrates in the metabolic network. Conc., concentration.

ion for δ*t* = 0 is shown in Supplementary Fig. 21). These compounds show varying degrees of mutual information with the three inputs, where several compounds, such as $[C_6H_{12}O_6Ca]^{2+}$ and $[C_7H_{14}O_7Na]^+$ highlighted in the figure, show increased mutual information with both past and future input (corresponding to negative and positive values of δ*t*), confirming that some compounds exhibit a short-term memory. We propose that these species are involved in reactions at longer timescales and can thus function as a type of memory. Formaldehyde shows much lower overall mutual information than the other two inputs. This may, again, be due to the low sensitivity of the formose reaction to changes of formaldehyde concentration on the timescale of the applied dynamics.

The heterogeneous network memory effects we observe are crucial to the operation of fully autonomous systems, which can anticipate and understand changes in their surrounding environment and respond accordingly. The chemical interface demonstrated here shows how we can intimately tap into complex molecular information from the environment, integrating and consolidating it into a well-defined response.

## Conclusion

We have demonstrated in chemico reservoir computing on the basis of information processing by a self-organized chemical reaction network.

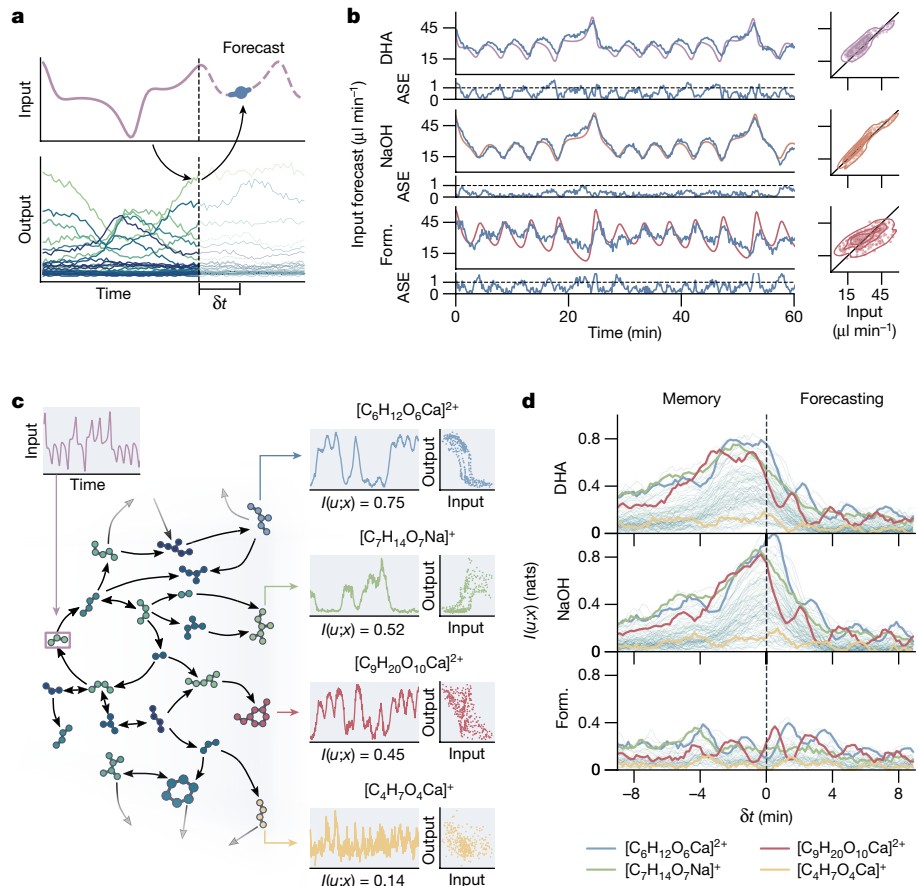

**Fig. 4 | Memory and prediction in the formose reservoir computer.**
**a**, Schematic of the prediction procedure to forecast inputs using the reservoir. A time-varying input is fed into the reservoir and the reservoir response recorded. Weights are trained on the reservoir state and the input at a time interval $\delta t$ into the future. These trained weights are then used to forecast as-yet unseen future inputs. **b**, Time traces, error plots and comparison plots for forecasts of simultaneously varying DHA, NaOH and formaldehyde inputs that resemble the behaviour of a Lorenz attractor. True inputs are shown as purple, orange and red lines, and the forecasts ($\delta t = 120$ s) as blue lines. The ASEs (see Methods) over time are shown below the predictions. **c**, A schematic showing how a time-dependent input propagates through the formose network, with different compounds responding in distinct ways. Only the DHA input is shown (left). The response over time of four ion signals is shown, as well as comparison plots between the DHA input and each output. Below every plot, the direct mutual information between DHA input and ion signal is shown ($I(u;x)$). **d**, A plot of the mutual information between ion signals $x(t + \delta t)$ and the formaldehyde, NaOH and DHA input patterns $u(t)$ as a function of the lag parameter $\delta t$. Four traces corresponding to the ion signals in **c** are indicated.

This system can perform several classification tasks in parallel, emulate the behaviour of biochemically important reaction networks and forecast changes in chaotic dynamical environments. These capabilities are reminiscent of how biological systems process and respond to environmental information, thus providing an interface through which autonomous systems, such as artificial cells or electronic devices, may receive and learn from the chemical environment.

Our approach circumvents some potential limitations in designed (bio)molecular computers, such as the need for explicit engineering of individual reactions and limited generalizability, giving rise to new opportunities in the development of chemical computers. The simple scalability and extensibility of self-organizing chemical reaction networks shows potential for rapid improvements, especially in large-scale computation and simulation of multiscale dynamical systems. In the future, the inclusion of different initiators, such as glycolaldehyde or erythrulose, could increase the number of available inputs into the system[4], and extension of the reaction with phosphorylated and cyanide-based compounds may further diversify complex reaction outputs[42,43].

A key challenge for in chemico computation is to replace the current electronic 'read-out layer' with a fully chemical read-out capable of autonomous learning. We provide an extra proof-of-concept experiment demonstrating how simple colorimetric read-outs of information processing may be implemented for the formose reaction

(Methods and Extended Data Fig. 5). By combining the reservoir output with selected reagents, a colorimetric response is produced that depends on both mixture composition and reagent, resulting in a specific hue or colour per input. The reagents thus function as a fixed read-out layer to the reaction mixture, chemically setting the read-out weights.

The information processing abilities of the formose reaction, and, potentially, of other self-organizing chemical networks, may offer a powerful interface with biological systems. For example, the formose reaction has been used to stimulate bioluminescent responses through interaction with a quorum-sensing pathway in the marine bacterium *Vibrio harveyi*[44]. Such an interface would allow us to establish a new class of intelligent matter, driven directly by the flux of information through chemical reaction networks. In a broader context, our work shows that complex chemical networks have inherent computational capabilities. By focusing on self-organization, complex molecular mixtures and nonlinearity, the full information processing capabilities of complex chemical systems may finally come to fruition.

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

## Methods

### Materials

$CaCl_2$, paraformaldehyde and tri-sodium citrate dihydrate were obtained from Sigma Aldrich. NaOH, thymol and copper(II) sulfate pentahydrate were obtained from Fisher Scientific. 1,3-Dihydroxyacetone dimer was obtained from Fluorochem EU. Resorcinol was obtained from TCI Europe NV. Formaldehyde solutions were prepared by depolymerization of paraformaldehyde at 60 °C; the final formaldehyde concentration was determined by titration with sodium sulfite and phenolphthalein[45]. For all aqueous solutions, ultrapure water, obtained from an Elga Purelab Chorus 1, was used. Before use, water was degassed by stirring under vacuum for 10–15 min. Ion mobility mass spectrometry experiments were performed with a timsToF instrument (Bruker Daltonics) equipped with an electrospray ionization source operating in positive mode.

### Flow reactions

A CSTR (volume 435 μl) with five inlets and an outlet was fabricated from poly(methyl methacrylate) by the Radboud TechnoCentre. LabM8 syringe pumps with BD Plastipak syringes were used to control input flow rates (a schematic drawing and a photograph of the set-up are provided in Extended Data Fig. 1). Syringes were loaded with the specified solutions and connected to the reactor with filled tubing. When the flow of water was divided between two syringes, the flows were fused using a Y connector before reaching the reactor. The reactor and a small outlet tubing were filled according to the initial conditions of the experiment. Once filled, the outlet tubing was capped with a one-way flow check valve and the system was allowed to build up pressure to overcome the crack pressure of the valve. Subsequently, the reactor output was diluted with a water flow (0.8 ml min⁻¹) controlled by a Bruker Elute HPG 1300 high-performance liquid chromatography system. The dilution flow was merged with the reactor outlet with a Y connector. With a subsequent Y connector, the flow was diverted between the instrument and a Restek RT-25020 backpressure regulator connected to a waste line. The backpressure regulator provided a constant pressure of 2 bars in the reservoir.

Inputs to the reactor were controlled by changing the flow rates of selected syringes. For a desired input concentration $C_{in}$, the flow rate can be calculated as $F = F_{tot} C_{in}/C_{syr}$, with $F_{tot}$ the total flow rate of the system (217.5 μl min⁻¹ in all experiments, corresponding to a residence time of 2 min) and $C_{syr}$ the concentration of the selected syringe.

### Flow inputs

Experimental conditions were selected based on previously published research[4], to create high compositional diversity over the used concentration range. The general workflow consisted of first generating a desired input function $u(t)$, and then scaling the generated function to a suitable input profile. To do so, the function was first mean-centred, scaled with a manually chosen factor, which was chosen to maximize amplitude without generating negative flows. After scaling, a baseline value of the corresponding syringe was added, so the profile fluctuated around the initial input concentration. The flow rate of water was used to counterbalance changes in flow rate, to ensure a constant total flow rate of 217.5 μl min⁻¹, corresponding to a residence time of 2 min. All reactions were allowed to equilibrate at steady state for at least 30 min before starting flow profiles, which included another initial 30 min of steady state. Details of the parameters used for generating the various flow profiles are provided in the respective Methods sections.

### Mass spectrometry

Trapped ion mobility spectrometry (TIMS) experiments were performed using an $N_2$ carrier gas by scanning inverse ion mobilities from 0.4 Vs cm⁻² to 0.84 Vs cm⁻². The ramp time was set to 500 ms and the accumulation time to 20 ms to minimize ion activation in the TIMS region. The mass range scanned by the time-of-flight (ToF) analyser was set to $m/z$ 50–650. A complete description of the instrumental parameters is available in Supplementary Information section 1.1.

### Ion intensity extraction

A list of ions with reference $m/z$ and inverse mobilities was established based on the most intense signals observed (Supplementary Information section 1.2). Ion chromatograms were then extracted for mass- and mobility-selected ions based on the reference list of ions using the TimsPy library[46]. Ion chromatograms were extracted with a mass width of 0.02 Da and a mobility width of 0.006 Vs cm⁻².

### Nonlinear classification

Varying input concentrations of formaldehyde and NaOH were applied with constant concentrations of DHA (50 mM) and $CaCl_2$ (15 mM). For every input in the nonlinear classification dataset, ion signals were collected for 30 min (Supplementary Information section 3.1 and Supplementary Figs. 5 and 6). The output in the last 10 min of this period were averaged to reduce noise and used as steady-state data, resulting in 106-dimensional vectors for all 132 inputs. These vectors were subsequently normalized to remove the mean and scaled to unit variance across features. For the selected nonlinear classification tasks, a linear support vector classifier was trained to obtain classifications of the inputs. For every task, a stratified leave-five-out cross-validation was performed, with 520 repeats in total, with every input as part of the test set 20 times (20 repeats of 26 random splits, five inputs per split), and the $\Phi$ score was calculated over the test set for every repeat as

$$\Phi = \frac{TP \times TN - FP \times FN}{\sqrt{(TP + FP)(TP + FN)(TN + FP)(TN + FN)}}$$

where TP denotes the number of true positives, TN denotes true negatives, FP denotes false positives and FN denotes false negatives. This score returns +1 for perfect predictions, and −1 for completely wrong predictions. The reported $\Phi$ accuracy was then obtained as $(\Phi + 1)/2$, and averaged over all 520 repeats. More information is available in Supplementary Information sections 3.2–3.6, and code is provided in the analysis/classification.ipynb notebook.

### Complex dynamics prediction

A fluctuating DHA flow profile was sampled from a normal distribution with a mean of 36.25 μl min⁻¹ and a standard deviation of 10.36 μl min⁻¹, corresponding to a mean input concentration of 50 mM, with a standard deviation of approximately 14.268 mM. Each flow rate was held constant for 60 s before switching, with an inversely fluctuating water input to ensure the total flow rate remained constant. The formaldehyde, NaOH and $CaCl_2$ inputs were held constant at 50 mM, 30 mM and 15 mM, respectively.

For the in silico simulation of the carbon metabolism of *E. coli*, a Systems Biology Markup Language (SBML) model from ref. 36 was adapted to include inflow and outflow terms for every substrate of the form $\varnothing \to X (k_f X_{in})$ and $X \to \varnothing (k_f X)$. The flow constant (residence time) was set to $k_f = 0.5$ min⁻¹, and the inflow concentrations $X_{in}$ were set to the initial concentrations of the model. This modified SBML file was subsequently compiled into a C++ module by the AMICI computational package[47] and loaded as a Python module.

To generate the training and test sets, the model was first run for 1,000 min until a steady state was reached. Then, for every step in the fluctuating input pattern, the DHA input flow concentration was set to the corresponding value of the physical fluctuating flow profile. The model was simulated with this input flow for the duration of the physical flow profile (1 min) before the new DHA input flow was set. For every step, the simulation was initialized at the final state of the previous step. By appending the results of all simulation steps, a complete record of the behaviour of the network under fluctuating conditions was obtained.

Next, the same DHA input flow was used as input into the formose reservoir. The response of the formose reservoir was collected every 500 ms, after which the output was averaged over bins of 10 s to reduce noise. The recorded formose reservoir response was trained on the individual substrate time series of the model for 30 min, using a ridge regression algorithm with the regularization strength set to $\alpha = 5 \times 10^{-5}$. The trained weights were then used to predict the substrate time series directly from the reservoir output for the remainder of the measurement time (code available in the analysis/dynamics.ipynb notebook).

### Absolute scaled error
To compare predictions for the dynamic tasks to the true values over time, we calculated the absolute scaled error (ASE), which is the absolute error between predictions and true values, divided by the mean absolute error of a naive mean forecast based on the training data. This error measure produces scale-invariant values that can be used to compare predictions across different data scales.

$$\mathrm{ASE}(t) = \frac{|\hat{y}(t) - y(t)|}{\frac{1}{T_{\mathrm{train}}} \sum_t^{T_{\mathrm{train}}} |y(t) - \bar{y}_{\mathrm{train}}|}$$

where $\hat{y}(t)$ is the prediction at time $t$, $y(t)$ is the true value at time $t$ and $\bar{y}_{\mathrm{train}}$ is the mean value of the train set.

### Forecasting
DHA, NaOH and formaldehyde inputs were simultaneously varied according to the dynamics of a Lorenz attractor ($\rho = 28$, $\sigma = 10$, $\beta = 8/3$) over the duration of 8 h, with a constant concentration of $CaCl_2$ (15 mM). The $x$, $y$ and $z$ axes were scaled to the NaOH, DHA and formaldehyde inputs by 1.4, 1.0 and 1.3, respectively. For a more detailed description of the flow profile used, see Supplementary Information section 5.1. The reservoir response was measured every 500 ms, after which the output was averaged over bins of 10 s to reduce noise. Next, a ridge regression algorithm was used with the regularization strength set to $\alpha = 5 \times 10^{-5}$ to train the formose response on the input flows 2 min (120 s) into the future for a duration of 30 min. The trained weights were then used to forecast the input flows 2 min into the future directly from the reservoir output for the remainder of the measurement time (code available in the analysis/forecast.ipynb notebook).

### Mutual information
Mutual information is defined for a pair of random variables $X$ and $Y$ as

$$I(X;Y) = \sum_Y \sum_X P_{(X,Y)}(x,y) \log\left(\frac{P_{(X,Y)}(x,y)}{P_X(x)P_Y(y)}\right)$$

with $P_X$ and $P_Y$ the marginal distributions, and $P_{(X,Y)}$ the joint distribution of the random variables. This formula was adapted to calculate the mutual information between a time-dependent input signal $u(t)$ and a single ion output signal at a different time $x(t + \delta t)$ as

$$I_{U;X}(\delta t) = \sum_t P_{U,X}(x_{t+\delta t}, u_t) \ln \frac{P_{U,X}(x_{t+\delta t}, u_t)}{P_X(x_{t+\delta t})P_U(u_t)} \tag{1}$$

where $u(t)$ is a time-dependent input flow and $x(t)$ is a single ion trace over time. An implementation from the Scikit-learn computational package[48,49] was used to perform the calculations (code available in the analysis/mutual_information.ipynb notebook).

### Chemical read-out for the formose reaction
Benedict's reagent was prepared by dissolving sodium citrate (8.65 g) and $Na_2CO_3$ (5.0 g) in 40 ml of water. $CuSO_4 \cdot 5H_2O$ was dissolved in 5 ml of water and slowly mixed with the sodium citrate, sodium carbonate solution. The solution was further diluted to 50 ml total volume. Seliwanoff's reagent was prepared by dissolving resorcinol (25 mg)

or thymol (25 mg) in HCl (3 M, 50 ml). Polytetrafluoroethylene (PTFE) tubing, 1/16" outside diameter × 0.032" inside diameter, was used as a plug-flow reactor, with a total volume of 480 µl. Unless otherwise specified, concentrations were 50 mM for DHA, 30 mM for NaOH, 15 mM for $CaCl_2$ and 100 mM for formaldehyde, with a 4 min residence time. Output was sampled by connecting the plug-flow reactor to a Bio-Rad drop former. Droplets were collected in a liquid nitrogen-cooled microplate, with two droplets (70 µl) per well. To each well, 150 µl of colorimetric reagent (Benedict's, Seliwanoff's resorcinol or Seliwanoff's thymol) was added. The microplates were heated to 100 °C using a Grant-bio PHMP-100, and pictures were taken at regular intervals, using a Nikon Z5 with a Laowa 100-mm F2.8 CA-Dreamer Macro 2X. Images were adapted to plots using OpenCV-Python[50].

To achieve a direct read-out of the formose reservoir, we added reagents to the reaction mixture. The overall sum of the concentration of compounds in the mixture 'multiplied' by the reaction with the added reagent results in a specific colorimetric response depending on both mixture composition and reagent. As Extended Data Fig. 5 shows, each combination results in a specific hue or colour. The reagents thus function as a fixed read-out layer to the reaction mixture, chemically setting the read-out weights. The reagents tested here have different mechanisms for their colorimetric response: Benedict's reagent functions under basic conditions, through the oxidation of Cu(II) to Cu(I), and the associated colour changes from blue to red; Seliwanoff's reagent (both for resorcinol and thymol) functions under acidic conditions, where compounds are first dehydrated towards furfural derivates, followed by a condensation reaction with resorcinol or thymol, forming a coloured dye. Seliwanoff's reagent classically uses resorcinol, but, in principle, other phenolic compounds can be used. We demonstrate colorimetric responses for a grid consisting of varying DHA and NaOH inputs. For Benedict's reagent, we observe increasing sensitivity with decreasing NaOH:DHA, compared to Seliwanoff's reagent, for which we observe increasing sensitivity with increasing NaOH:DHA for both the resorcinol- and the thymol-based reagents.

The chemical read-out allows for the identification of different environmental inputs, especially if combinations of different reagents or reaction times are used. Potentially, this approach may be further extended to incorporate a feedback mechanism that can change the amount and type of reagents and modify other system hyperparameters to perform a specific computational task, either through the inclusion of an in-the-loop computer or directly through physical learning[51].

## Data availability
Data are available at Zenodo (https://doi.org/10.5281/zenodo.10136537)[52].

## Code availability
Python code for working with the datasets, as described in the Methods and Supplementary Information, is available at Zenodo (https://doi.org/10.5281/zenodo.10136537)[52].

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

**Acknowledgements** We thank B. van Sluijs and O. Maguire for helpful discussions and suggestions. We thank M. Derks (Labm8) for his help and work in the design of the syringe pumps and flow set-up. We thank the Radboud TechnoCentre, specifically A. de Kleine, for the design and manufacturing of the reactors. This project has received funding from the European Union's Horizon 2020 research and innovation programmes under grant agreement no. 833466 (ERC Advanced Grant Life-Inspired), the Dutch Ministry of Education, Culture and Science (Functional Molecular Systems, Gravitation programme 024.001.035), the Dutch Research Council (grant OCENW.KLEIN.348) and the Simons Collaboration on the Origins of Life (SCOL; award 477123), and was also supported by the Radboud–Glasgow Collaboration Fund, and the European Union and the Swiss State Secretariat for Education, Research and Innovation (SERI) under contract numbers 22.00017 and 22.00034 (Horizon Europe Research and Innovation Project CORENET).

**Author contributions** M.G.B. and W.T.S.H. conceived the research. T.J.d.J., Q.D. and W.E.R. developed the experimental methodology. M.G.B., T.J.d.J. and W.E.R. designed the experiments, and T.J.d.J. and Q.D. performed the experiments and preprocessed the data. M.G.B., T.J.d.J., Q.D. and W.E.R. wrote the data analysis software. M.G.B. performed the analysis and trained the models. W.T.S.H. and W.E.R. supervised the project. M.G.B. and W.T.S.H. wrote the original draft, and all authors discussed results and contributed to writing the manuscript.

**Competing interests** The authors declare no competing interests.

## Additional information

**Correspondence and requests for materials** should be addressed to Wilhelm T. S. Huck.

**a**

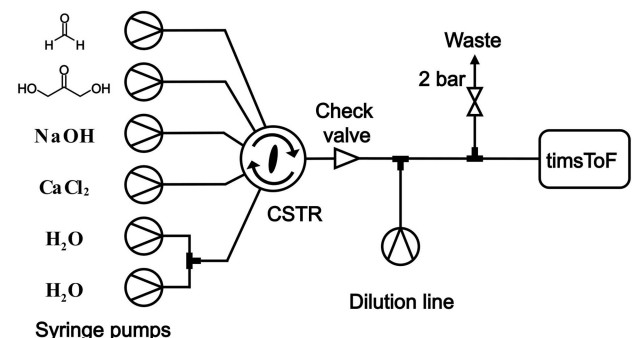

**b**

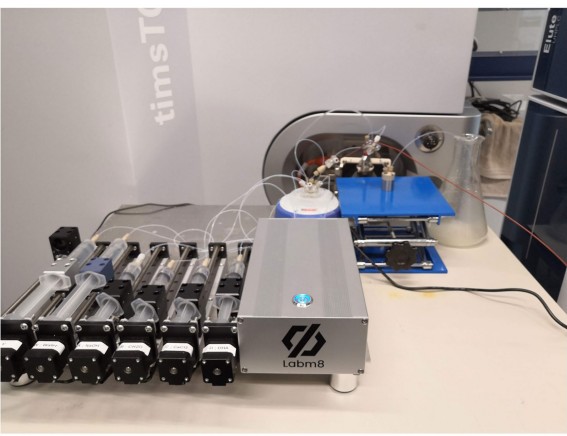

**Extended Data Fig. 1 | Experimental setup.** a) Schematic overview of the flow reactor setup. Syringes are mounted inside syringe pumps and fed into the CSTR. One way flow from the reactor is ensured by a check valve. Flow is subsequently diluted and split to waste and timsToF using a back-pressure regulator. b) Photograph of the experimental setup, syringe pumps are in the bottom left, the CSTR is on top of the white blue stirring plate.

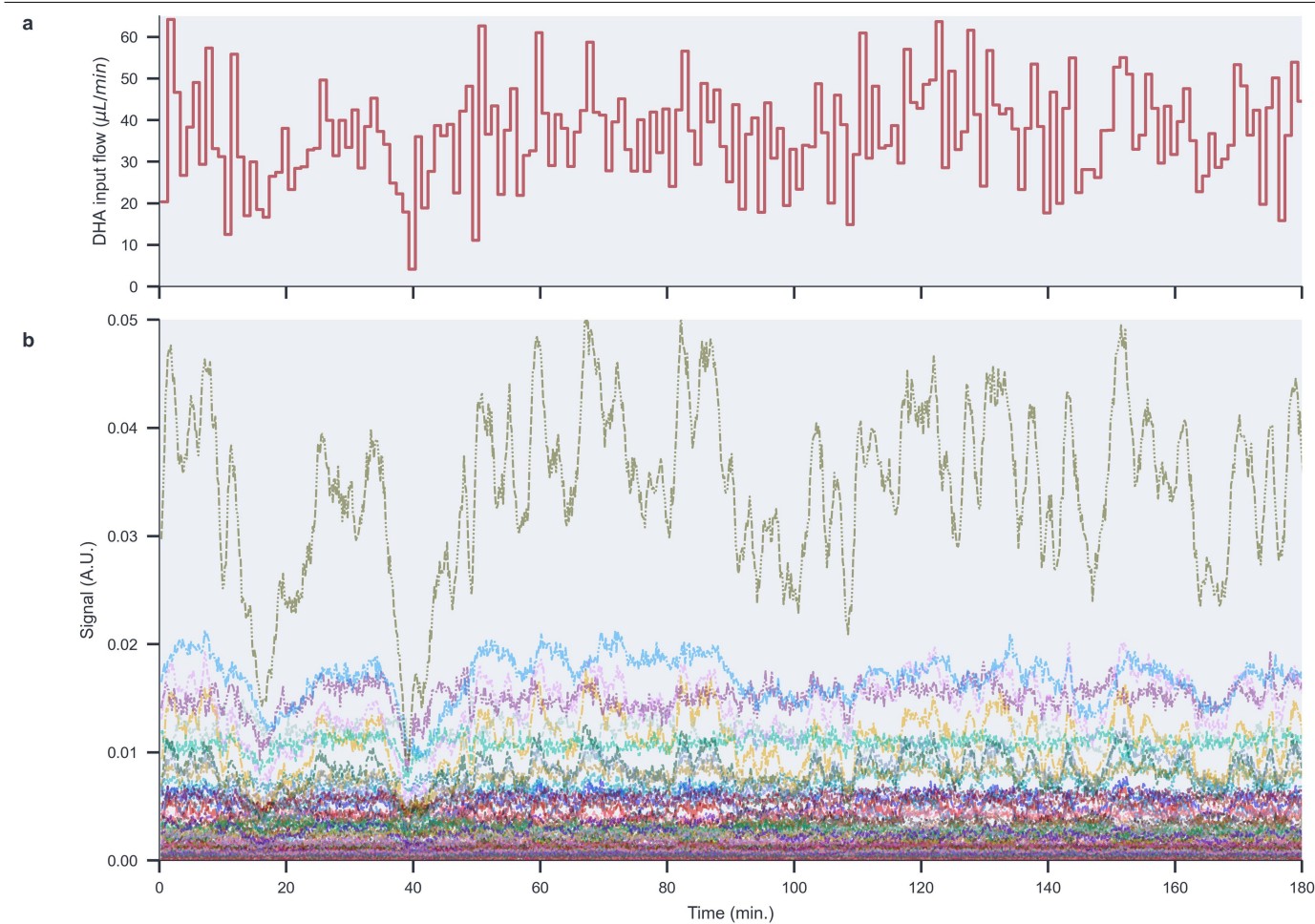

**Extended Data Fig. 2 | Fluctuating input flow and reservoir response.**
a) Fluctuating input flow profiles used in the prediction of metabolic network behaviour. CaCl$_2$, formaldehyde, and NaOH were kept constant at a flow rate of 36.25 µL/min, while DHA was varied. b) Ion signals observed in response to the changing flow inputs. Colours indicate different ion signals.

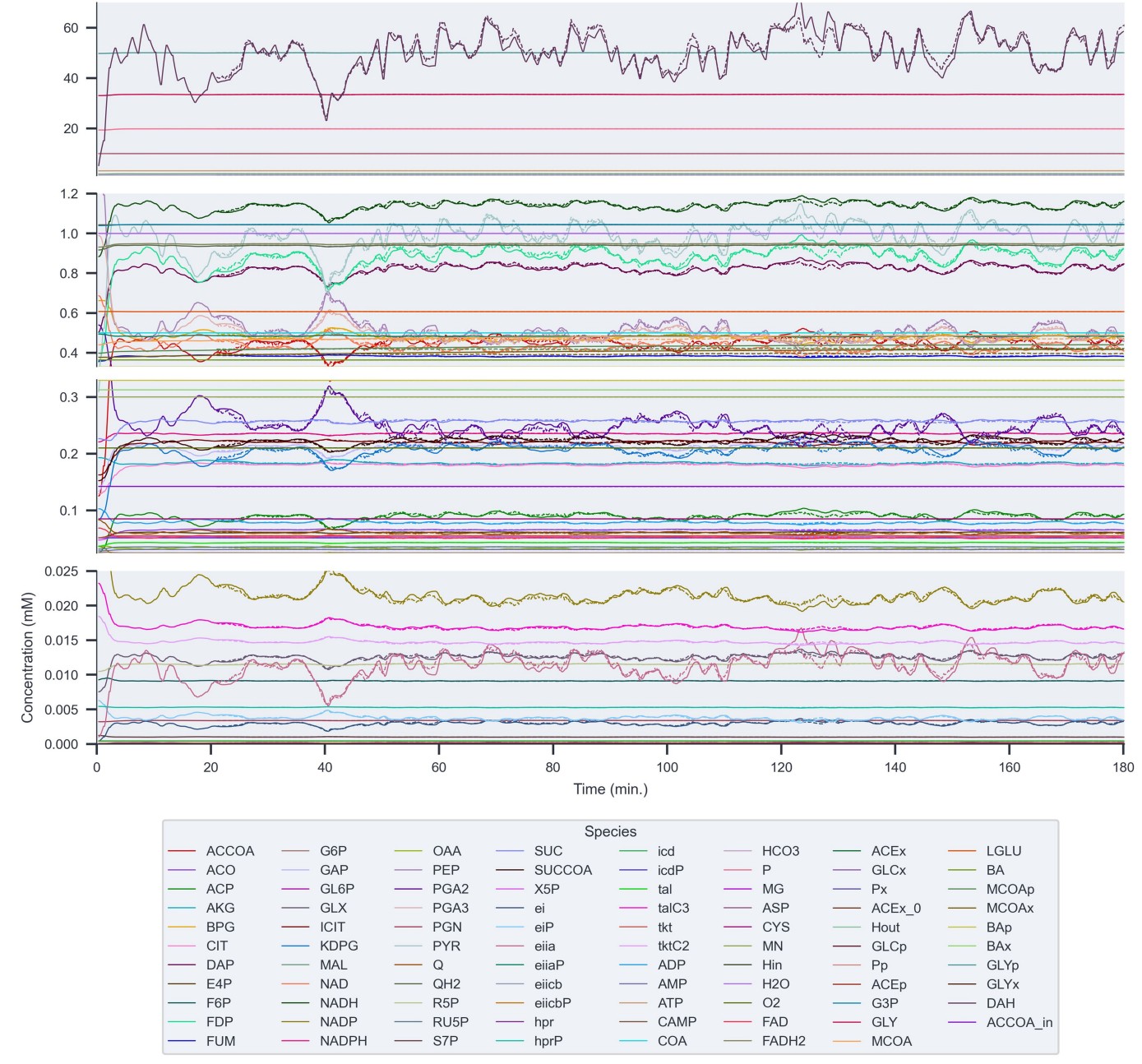

**Extended Data Fig. 3 | Prediction results for substrates in the metabolic network.** Prediction results for substrates in the metabolic network. True (simulated) time series are shown as solid lines, predictions of the trained formose reservoir as dashed lines. Four different substrate concentration regimes are shown.

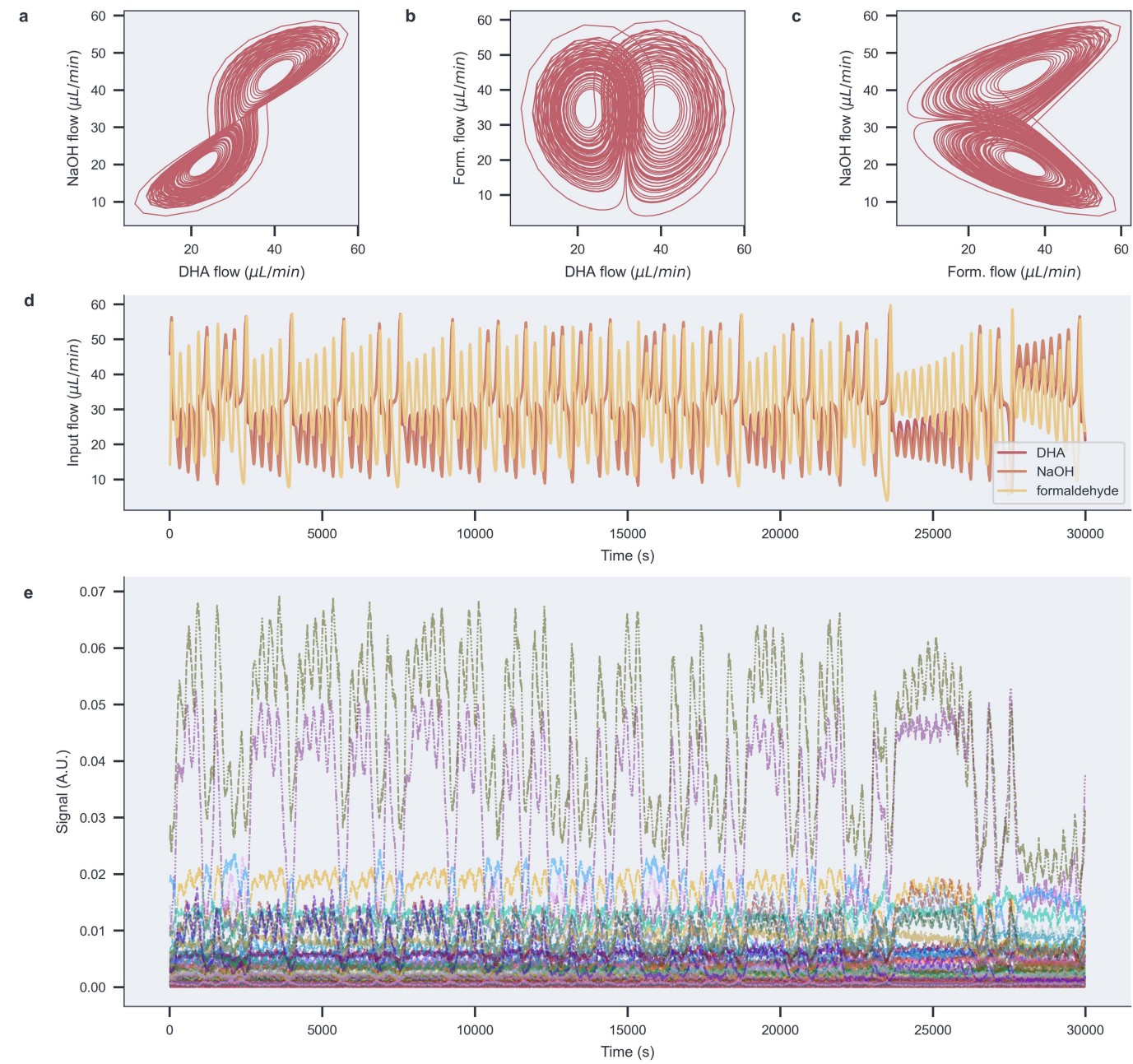

**Extended Data Fig. 4 | Lorenz attractor input flow and reservoir response.**
a-c) Orthogonal projections of the Lorenz attractor on the input space, represented by the flow rates of three reaction inputs (DHA, NaOH and formaldehyde) d) Dynamic flow profiles for DHA, NaOH, formaldehyde, and water for the Lorenz attractor experiment. The flow of $CaCl_2$ was kept constant at a rate of 30.2083 µL/min. e) Ion signals observed in response to the changing flow inputs. Colours indicate different ion signals.

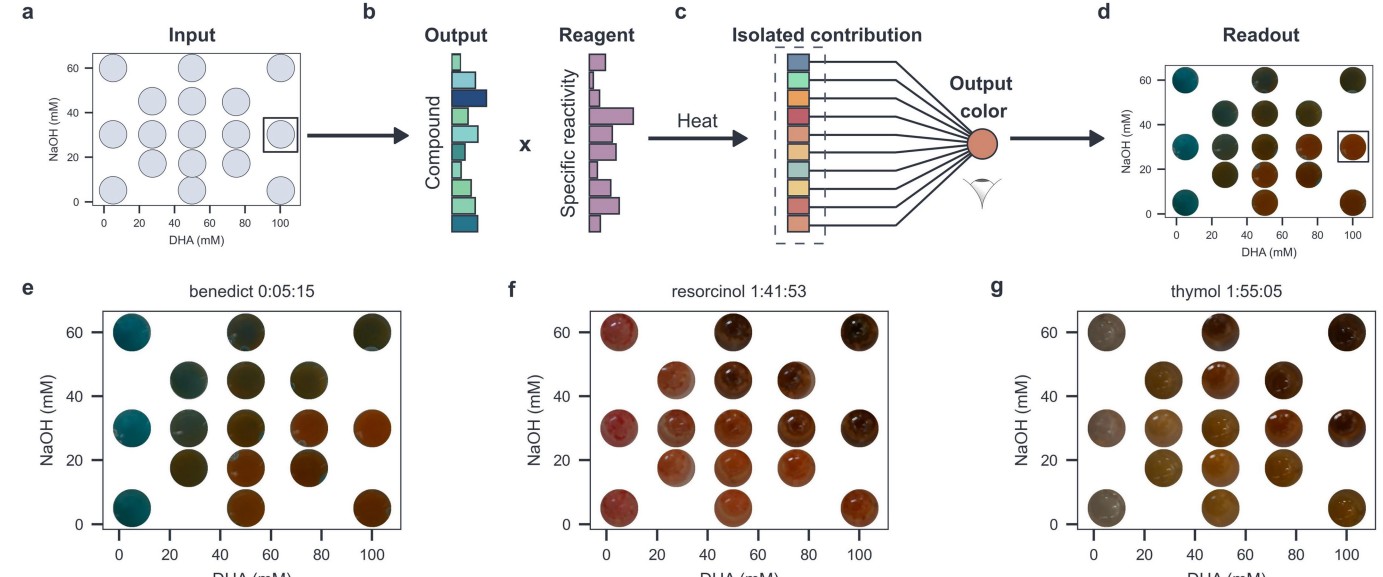

**Extended Data Fig. 5 | Schematic overview and results of classification using colorimetric readout.** a) A scatter plot showing the sampled DHA and NaOH b) each sample consists of a unique composition of compounds, depending on the conditions each compound has a specific reactivity with the reagent, this can also be viewed as a set of c) In isolation each compound would produce a different color, depending on its concentration and reactivity with the reagent. These isolated contributions cannot be observed, instead we observe one final output colour that can be considered a sum of the individual effects. d) A potential final readout. e) Result of visual readout using Benedict's reagent after 5 min. f) Result of colorimetric test using Seliwanoff's resorcinol reagent after 1 h and 42 min. e) Result of colorimetric test using Seliwanoff's thymol reagent after 1 h and 55 min.

**Extended Data Table 1 | Classification accuracies**

| Task | FRC | LSVC | SVC | GP | MLP | ELM |
|---|---|---|---|---|---|---|
| AND | 0.97 | 0.93 | 0.94 | 0.86 | 0.95 | 0.94 |
| OR | 0.90 | 0.90 | 0.95 | 0.89 | 0.89 | 0.95 |
| Linear | 0.91 | 0.94 | 0.92 | 0.93 | 0.95 | 0.93 |
| Triangle | 0.94 | 0.97 | 0.98 | 0.96 | 0.60 | 0.97 |
| XOR | 0.88 | 0.51 | 0.92 | 0.72 | 0.96 | 0.93 |
| Checkers | 0.68 | 0.50 | 0.67 | 0.50 | 0.50 | 0.42 |
| Circle | 0.88 | 0.54 | 0.99 | 0.71 | 0.97 | 0.95 |
| Sine | 0.87 | 0.67 | 0.94 | 0.67 | 0.87 | 0.72 |
| Concentric | 0.77 | 0.41 | 0.82 | 0.54 | 0.71 | 0.83 |
| Dots | 0.83 | 0.45 | 0.90 | 0.48 | 0.86 | 0.83 |

Numerical values for the average test-set $\Phi$-accuracy for 520 different leave-5-out train-test splits (Methods). Values are reported per task as seen in Fig. 1 of the main text; for the formose reservoir (FRC), training layer without reservoir (LSVC: Linear Support Vector Classifier), and various machine learning classifiers (SVC: Support Vector Classifier, GP: Gaussian Process Classifier, MLP: Multilayer Perceptron, ELM: Extreme Learning Machine). A score of 1.0 corresponds to perfect predictions, and a score of 0.0 to total failure. A score of 0.5 equals random guesses taking into account classification group sizes.