## [Peer Review File · Nature]

Editorial Note: Figure 3 on Page 12 in this Peer Review File is reprinted (adapted) with permission from Liu, X. and Parhi, K.K. Reservoir Computing Using DNA Oscillators. ACS Synthetic Biology, 11(2), 780-787 (2022). Copyright 2022 American Chemical Society.

Manuscript Title: Chemical reservoir computation in a self-organizing reaction network **Reviewer**

Comments & Author Rebuttals

Reviewer Reports on the Initial Version:

Referees' comments:

Referee #1 (Remarks to the Author):

The main contribution of the paper is the implementation of a chemical reservoir computer based on the formose reaction. To the best of my knowledge, this has not been done before. The work is also significant because it is based on an actual chemical realization of the reservoir and not just based on numerical simulations. However, a key part of the system, the readout layer, is not implemented in chemistry. I'd argue that a "major breakthrough," as the authors claim, would consist in a fully autonomous chemical implementation. While not trivial, it's still rather straightforward to use almost any physical system as a reservoir. The Cucci et al. paper, which the authors cite, shows nicely how to do just that. So, arguing that using a chemical reservoir presents a "major breakthrough" seems a little far-fetched.

In-situ learning for simple chemical networks was previously proposed. However, in chemico implementation was ever done, as far as I know:

(1) Lakin, M.R., 2023. Design and Simulation of a Multilayer Chemical Neural Network That Learns via Backpropagation. Artificial Life, pp.1-28.

(2) Drew Blount, Peter Banda, Christof Teuscher, Darko Stefanovic; Feedforward Chemical Neural Network: An In Silico Chemical System That Learns xor. Artif Life 2017; 23 (3): 295–317. doi: https://doi.org/10.1162/ARTL_a_00233

If an in chemico implementation based on formose reactions is not currently technical feasible, the authors could perhaps propose at least a numerical simulation and/or proof-of-concept? I think this is really what this paper actually needs because there just isn't much novelty in demonstrating that a given chemical/physical/biological system can be used as a reservoir.

Reservoir computing with self-organized chemical systems is also not quite new, as the citations below illustrate:

(1) Goudarzi, A., Lakin, M.R. and Stefanovic, D., 2013, September. DNA reservoir computing: a novel molecular computing approach. In International Workshop on DNA-Based Computers (pp. 76-89). Cham: Springer International Publishing.

(2) Liu, X. and Parhi, K.K., 2022. Reservoir Computing Using DNA Oscillators. ACS Synthetic Biology, 11(2), pp.780-787.

(3) Yahiro, W., Aubert-Kato, N. and Hagiya, M., 2018, July. A reservoir computing approach for molecular computing. In *Artificial Life Conference Proceedings* (pp. 31-38). One Rogers Street, Cambridge, MA 02142-1209, USA journals-info@mit.edu: MIT Press.

(4) Nguyen, H., Banda, P., Stefanovic, D. and Teuscher, C., 2020, July. Reservoir computing with random chemical systems. In *Artificial Life Conference Proceedings 32* (pp. 491-499). One Rogers Street, Cambridge, MA 02142-1209, USA journals-info@mit.edu: MIT Press.

To improve the manuscript, I'd suggest the authors provide a direct comparison with the chemical reservoirs that were previously published, including the same tasks.

Several times the authors point to the fact that their implementation demonstrates that biological and chemical systems can perform a variety of computational tasks. I don't think this should come as a surprise to anyone. I also don't think it's surprising that the formose reaction can be used to perform complex information processing because almost any physical system can do so when used as a reservoir. The bar for "complex information processing" (whatever that actually means) is really pretty low when it comes to reservoir computing.

Quantifying how information propagation through the system was a neat addition, not usually provided by the physical reservoir computing community. However, a key question that remains unexplored: what are the computational limits of a formose reaction reservoir? How complex of a task can one solve? And how would you scale up such a system to solve more complex tasks? The authors mention the potential for large-scale computation, but it is unclear how one would get there with their approach. Perhaps the authors could try to deduce an upper limit by using their mutual information approach?

As far as I can tell, the methodology is solid and appropriate methods and data is used. I also found the paper to be very well written. The illustrations are clear and support the paper's narrative.

Referee #2 (Remarks to the Author):

A.

The manuscript „Chemical reservoir computation in a self-organizing reaction network” by Mathieu G. Baltussen, Thijs J. de Jong, Quentin Duez, William E. Robinson, and Wilhelm T.S. Huck describes experimental results demonstrating computing with formose reaction. The authors consider 5 reagents that can be used for data input and measure up to 106 ion concentrations to obtain the time evolution of the reservoir. After the measurements, the concentrations are processed by a single artificial neuron to produce the output. To demonstrate the computing potential of the considered medium, the authors consider such problems as the classification of points belonging to a specific 2-dimensional structure, simulations of a dynamical system, and the prediction of time evolution based on previously collected knowledge.

B.

In my opinion, the manuscript can be interesting for a broad spectrum of readers, and it is appropriate for publication in Nature because it shows the experimental realization of a chemical computer that can perform non-trivial operations.

C & F. & H.

I have some remarks and questions that should be resolved before the manuscript is accepted.

Through the manuscript, the authors call the network self-organizing. Could they explain what this self-organization is? In my opinion, the reactions are just determined by the presence of specific reagents in the system.

All classification problems used the same set of 132 randomly sampled concentrations of formaldehyde and NaOH. I presume not all points of the test set were classified correctly because always the Φ -coefficient is <1 .

- Could the authors mark correctly and incorrectly classified points in Fig.2 (for example, using different colors as in Entropy, 2022, 24(8), 1054)?
- Have the input data different from those included in the training dataset been used for independent verification of the classifier accuracy?
- It would help if the authors give numbers of both Φ -coefficient and the accuracy $(= (1+\Phi)/2)$ for all FRC classifiers.
- What were the radii of circles in Circle, Concentric, and Dots problems?
- Could the authors explain the significant difference between the accuracy of AND and OR classifiers? The problems are isomorphic, so it is probably related to a small training sample.

As an example of modeling complex dynamics, the authors demonstrate that the formose reaction can replicate the time evolution calculated using a complex reaction model. The authors use fluctuating DHA as the input because it is the common reagent in both systems. On the basis of these results, they propose the mapping between the measured concentrations in the formose reaction and the calculated concentrations in the E.coli model. A nice match has been observed, so the authors claim the chemical reservoir can compute the model.

But how useful is such a simulator? The authors say,, by continuing the fluctuating input pattern after the training period, the learned linear mapping allows to use the formose reservoir as an emulator of the dynamic system ,, . What does the fluctuating $u(t)$ mean for times > 30 min? Do the fluctuations repeat the previous pattern?

- Does the agreement hold if one considers another $u(t)$ with the same average and dispersion?
- Is there any use of the simulator trained with fluctuating $u(t)$ when the input is constant and close to the average?

In my opinion, the part on forecasting applications of the reservoir computer is not clearly presented. According to previously discussed applications, $u(t)$ represents the reagent inflow, and it is a free parameter that can be modified at any time according to the experimentalist's wish. Therefore, there is no argument to expect that the relationship in the form $u(t+dt) = W x(t)$ is valid in general. The situation changes if the function $u(t)$ is fixed before the experiment, and we want to reproduce it from system observations in a time sub-interval shorter than the whole experiment.

Our prediction should be accurate if $u(t)$ is periodic and the sub-interval covers the whole period. It leads to complete failure if the selected sub-interval does not cover all representative parts of $u(t)$. In the considered example, the authors assumed that the Lorenz attractor describes changes in $u(t)$, and they focused their attention on restoring this input.

In my opinion, the introduction to forecasting applications that includes the information given above would help the readers to get the idea of forecasting (of course, if I have got the idea right).

G. OK

Referee #3 (Remarks to the Author):

The manuscript experimentally demonstrates physical reservoir computation using chemical reaction networks related to formose reactions (which are found in synthesizing sugar from formaldehyde). The capability of the chemical reaction network as a physical reservoir is demonstrated for three machine learning information processing tasks, including (i) nonlinear classifications of linearly inseparable data in two classes, (ii) dynamical system approximations for metabolic models, and (iii) autonomous dynamical system emulations with the Lorenz attractor exhibiting chaotic behavior.

This work is directly along the concept of physical reservoir computing which enable machine learning computation by using an untrainable physically-implemented dynamic reservoir and a trainable readout (Tanaka et al., Recent advances in physical reservoir computing: A review, *Neural Networks*, 2019). The idea of using chemical and electrochemical reaction systems as a dynamic reservoir is not so novel as there exist some previous studies (e.g. Kan et al., Physical Implementation of Reservoir Computing through Electrochemical Reaction, *Advanced Sciences*, 2021; Nguyen et al., Reservoir Computing with Random Chemical Systems, *Artificial Life Conference Proceedings*, 2020). It is not surprising to imagine that complex chemical reaction networks as used in the current work may be available as a dynamic reservoir after seeing the previous studies on In-Materio reservoir computing (Dale et al., Reservoir Computing as a Model for In-Materio Computing, *Advances in Unconventional Computing*, 2016). In the reservoir computing field, the impact of the current work is not so high under the existence of a plenty of implementation demonstrations of other types of physical reservoirs (e.g. Torrejon, J. et al. Neuromorphic computing with nanoscale spintronic oscillators. *Nature* 547, 428–431, 2017).

Nevertheless, as far as I know, this work is the first attempt to use the formose reaction network as a dynamic reservoir, presenting the high potential of chemical reservoir computing based on exhaustive experiments. The authors' sophisticated experimental system operating the self-organized chemical reactions (probably developed in their previous study [6]) seems to be the strong point of this work, enabling the inputs of various temporal signals, the control of chemical reaction processes, and the recording of time-varying chemical compound concentrations.

Considering that the concept of reservoir computing, the explanation in the schematic overview in Figure 1 is misleading. Normally a dynamic reservoir receives (time-)sequential input signals and reacts to a history of the input signals (not only to the current input signal), and therefore a reservoir computing system is suited for approximating a dynamical system. It is not mathematically correct to state that the target (Figure 1e) is a “function” of the input variable (Figure 1b). Correctly, the target output sequence is a transformation of the input sequence by a “filter” or a “dynamic system”. This is clearly described in the seminal papers of reservoir computing (Jaeger, The "echo state" approach to analysing and training recurrent neural networks-with an erratum note', GMD Technical Report, 2001; Maass et al., Real-time computing without stable states: A new framework for neural computation based on perturbations, Neural Computation, 2002). A correction of the corresponding description is necessary for explaining why the dynamical system emulation and forecasting (i.e. 2nd and 3rd tasks) are successful with the chemical “dynamic” reservoir. It should be noted that the classification of the static data (i.e. 1st task) shown in Figure 2 is a special case where the correspondence between an initial state and an equilibrium state of the reservoir is used as a static “function”. In addition, it does not make sense to compare the proposed method with the ESN (Echo State Network) because the classification performance of the ESN depends on the order of samples given to the reservoir but the order is meaningless in this task. My concern is that the authors do not care about the (mathematical) difference between an approximation of a “static function” and that of a “dynamic filter”.

The future perspective of this work is not fully understandable. It is reasonable that the chemical reaction networks are compatible with biological networks. However, the readout computation (i.e. an error minimization with a ridge regression) is currently done with a general-purpose computer. It is not clear how biological systems can implement the regression algorithms.

As for the practicality of the proposed method, I wonder if it is sufficient to test the chemical reservoir computation only with the artificial data generated by deterministic functions/dynamical systems. The capability of the proposed method for real data processing is still unclear.

Author Rebuttals to Initial Comments:

Referees' comments:

Referee #1 (Remarks to the Author):

The main contribution of the paper is the implementation of a chemical reservoir computer based on the formose reaction. To the best of my knowledge, this has not been done before. The work is also significant because it is based on an actual chemical realization of the reservoir and not just based on numerical simulations. However, a key part of the system, the readout layer, is not implemented in chemistry. I'd argue that a "major breakthrough," as the authors claim, would consist in a fully autonomous chemical implementation.

We thank the reviewer for their time in reviewing our manuscript and the helpful comments.

We agree with the reviewer that a fully autonomous chemical implementation of the readout layer including the training step would reinforce the power of this approach. However, we note also that the current dependence on *in silico* training is present in all fields of physical reservoir computation and is therefore a major open challenge on its own.

The reviewer's comments have stimulated us to demonstrate how a chemical readout layer could be implemented. We have performed proof-of-principle experiments on a grid of inputs using reagents which give a colorimetric readout. These reagents react with subsets of the chemical output. Specifically, Benedict's reagent reacts with reducing sugars, whereas thymol and resorcinol react with compounds capable of forming five-membered carbon rings. When combined with the formose reaction's output, they produce varying colour responses, which can be inspected visually. This process is in effect a readout layer, which sets weights on various compounds according to their reactivity and the colour produced is the effective sum the concentrations of compounds multiplied by their propensity to react with the reagent. These visible outputs could then be overlaid to produce a combined readout directly from chemical reactions (see figure 1 below). We have modified our conclusion to further explain the potential of such a chemical readout:

>>> Added/modified text in yellow (p15-16): "Importantly, while the reservoir computation paradigm is used in this work to establish and exploit the information processing capabilities of the formose reaction, in the future this electronic 'readout layer' may be replaced, or altogether omitted, by interfaces and reservoirs tailored to specific applications. So far, physical reservoir computers always depend on *in silico* training of the readout layer. A significant next step for *in chemico* computation would be a fully chemical readout capable of autonomous learning. Let us therefore present a first proof-of-concept experiment, in which we show how simple reagents that react with components of the reaction mixture, can be added to the output of the formose reservoir to chemically set readout weights. The overall sum of the concentration of compounds in the mixture 'multiplied' by the reaction with the added reagent results in a colorimetric response (see SI sections 1.3 and 3.7, and figure S19). As the figure shows, each combination of input parameters results in a specific hue or colour. Thus, the chemical readout allows for the

identification of different environmental inputs, certainly if combinations of different reagents or reaction times are used. Potentially, this approach may be further extended to incorporate a feedback mechanism that can change the amount and type of reagents and modify other system hyperparameters to perform a specific computational task, either via the inclusion of an in-the-loop computer or directly via physical learning[48]. The information processing abilities of the formose reaction, and potentially other self-organizing chemical networks, may offer a powerful interface with biological systems, allowing for direct chemical communication with cellular signalling pathways. In fact, the formose reaction has already been used to communicate with cells, stimulating bioluminescent responses via interaction with a quorum sensing pathway in the marine bacterium *Vibrio harveyi*[49]. The approach presented here could potentially be adapted to control such cellular responses in sophisticated manners by communicating with cells in their own 'language' via a direct chemical interface. Such an interface would allow us to establish a new class of intelligent matter, driven directly by the flux of information through chemical reaction networks."

Where we added the following reference:

48. Stern, M. & Murugan, A. Learning Without Neurons in Physical Systems. *Annual Review of Condensed Matter Physics* 14, 417–441 (2023).

Fig. 1 (Supporting Information figure S19): Schematic overview and results of classification using colorimetric reagents. a) A scatter plot showing the sampled DHA and NaOH concentrations. **b)** each sample consists of a unique composition of compounds, depending on the conditions each compound has a specific reactivity with the reagent, this can also be viewed as a set of weights. **c)** In isolation each compound would produce a different color, depending on its concentration and reactivity with the reagent. These isolated contributions cannot be observed, instead we observe one final output colour that can be considered a sum of the individual effects. **d)** A potential final readout. **e)** Result of visual readout using Benedict's reagent after 5 minutes. **f)** Result of colorimetric test using Seliwanoff's resorcinol reagent after 1 hour 42 minutes. **g)** Result of colorimetric test using Seliwanoff's thymol reagent after 1 hour 55 minutes.

Additionally, in the Supporting Information (figure S12), we have included an analysis of the nonlinear classification problems by training using Lasso regression. This method induces sparsity in the set of fitted weights (see figure 2 included below), thereby ‘selecting’ a more specific subset of compounds to be used in the prediction. These weights are directly attributed to the chemical compounds most relevant to each task. In principle, these specific compounds could be integrated into a singular output using a secondary chemical process. Figure 3 below further demonstrates how the reservoir can approximate an XOR gate if a sufficient subset of the compounds (selected by the Lasso regression) is observed.

Fig 2 (adapted from figure S12 in SI): Lasso regression allows for the selection of a specific subset of compounds which contribute to a classification task. Bars indicate the weights for each compound obtained from the Lasso regression training of an XOR gate classification task on a normalized data set. Compounds are numbered according to table 1 (Supporting Information).

Fig 3 (S13-17 in SI, shortened version included here): Visualization of the creation of an XOR gate by composition of different compounds with weights as obtained through a lasso regression (see

figure 2 above for all the weights), ordered from left-to-right by the absolute value of weights. Every fifth compound is shown here for brevity, a full version of 25 compounds is included in the Supporting Information. a) Row of scatter plots showing measured intensities for 5 different compounds (normalized values, compounds denoted by X_{xx}) and a shaded background obtained by interpolation. Plots are sorted by decreasing absolute weight value (e.g. the contribution in the classification process). Axis are the same as in manuscript figure 2, with NaOH input on the x-axis and formaldehyde input on the y-axis. b) For every compound in a), the measured intensities are multiplied by a weight value (denoted by W_{xx}) obtained from the lasso regression to obtain a weighted outputs for all 5 compounds. c) A set of five plots showing from left to right the sum of the weighted outputs for an increasing number of compounds (respectively 1, 6, 11, 16, and 21 compounds, ordered by absolute value of weight). By including more compounds in the summed output, an approximate XOR classification response is created.

While not trivial, it's still rather straightforward to use almost any physical system as a reservoir. The Cucci et al. paper, which the authors cite, shows nicely how to do just that.

Whilst the paper by Cucci et al. provides an excellent overview of how reservoir computation may be implemented in general, it provides no guidance on the actual chemistry, experimental methodology or analytical methods required to create a chemical reservoir computer. In this context, we cannot rely on literature findings that claim to demonstrate chemical reservoir computing. These reports detail *in silico* implementations which are based on unfeasible chemistry, or highly complex DNA-based systems. Consequently, no other chemical reservoir computers have been implemented physically until now.

So, arguing that using a chemical reservoir presents a "major breakthrough" seems a little far-fetched.

We respectfully disagree that our claim of a major breakthrough is far-fetched. Our work is a major breakthrough and represents the first experimentally realized reservoir computer based on a chemical network. Our novel approach, using complex self-organizing organic networks in flow in combination with sophisticated online-readout methods, circumvents limitations present in previously established molecular computation techniques. Thus, it represents a paradigm shift in approaches to molecular computation. We know of no other well-explored examples of reaction networks that possess the same inherent non-linear characteristics and computational capabilities. Furthermore, this network only exhibits reservoir computation capabilities in a carefully chosen regime of experimental conditions that was selected based on insight from previous work (in this case specific concentration regimes for Ca^{2+} , hydroxide, dihydroxyacetone and formaldehyde), highlighting that the molecular computation we describe really is something new and exciting, and far from trivial, as suggested by the reviewer.

In-situ learning for simple chemical networks was previously proposed. However, in chemical implementation was ever done, as far as I know:

(1) Lakin, M.R., 2023. Design and Simulation of a Multilayer Chemical Neural Network That Learns via Backpropagation. *Artificial Life*, pp.1-28.

(2) Drew Blount, Peter Banda, Christof Teuscher, Darko Stefanovic; Feedforward Chemical Neural Network: An In Silico Chemical System That Learns xor. *Artif Life* 2017; 23 (3): 295–317. doi: https://doi.org/10.1162/ARTL_a_00233

If an in chemico implementation based on formose reactions is not currently technical feasible, the authors could perhaps propose at least a numerical simulation and/or proof-of-concept? I think this is really what this paper actually needs because there just isn't much novelty in demonstrating that a given chemical/physical/biological system can be used as a reservoir.

We thank the reviewer for directing our attention to the above papers describing potential *in chemico* implementations of learning tasks. We would like to stress that the above-mentioned work is valuable in showing that *in chemico* learning is possible in theory, but these manuscripts do not implement learning in a physical chemical system, and do not suggest feasible experimental routes to implement autonomous learning. To better place our work in the context of these proof-of-concept studies, we have included additional statements in the introduction of the main text.

>>> Added/modified text in yellow (p2): “Significant progress has been made in constructing chemical systems that can act as logic gates and Boolean circuits[10,11], recreate digital computation architectures[12–14], artificial neurons and small neural networks capable of classification tasks[15-19], pattern recognition using the Belousov-Zhabotinsky reaction in combination with a deep convolutional neural network[20], **symbol recognition with chemical Turing machines[21]**, and general-purpose digital computation with DNA origami[22]. **Additionally, methods to create self-learning chemical systems have been proposed for abstract chemical reaction network models[23,24]**. Whilst these approaches demonstrate how molecular systems may be used to perform various computations, they do not achieve the information processing capabilities seen in living systems. **Furthermore, advancements have been limited both by constraints in representing digital processes through chemistry, and by the inherent increase in required experimental effort to scale up designed chemical systems.** Unlocking the potential of molecular systems requires: i) moving beyond a strict adherence to reproducing digital computation principles and ii) finding an approach which overcomes the laborious nature of bottom-up ‘molecule-by-molecule’ design patterns.”

Where we added the following references:

21. Dueñas-Díez, M. & Pérez-Mercader, J. How Chemistry Computes: Language Recognition by Non-Biochemical Chemical Automata. From Finite Automata to Turing Machines. *iScience* 19, 514–526 (2019).

23. Blount, D., Banda, P., Teuscher, C. & Stefanovic, D. Feedforward chemical neural network: An in silico chemical system that learns xor. *Artificial Life* 23, 295–317 (2017).

24. Lakin, M. R. Design and Simulation of a Multilayer Chemical Neural Network That Learns via Backpropagation. *Artificial Life* 29, 308–335 (2023).

Reservoir computing with self-organized chemical systems is also not quite new, as the citations below illustrate:

(1) Goudarzi, A., Lakin, M.R. and Stefanovic, D., 2013, September. DNA reservoir computing: a novel molecular computing approach. In *International Workshop on DNA-Based Computers* (pp. 76-89). Cham: Springer International Publishing.

(2) Liu, X. and Parhi, K.K., 2022. Reservoir Computing Using DNA Oscillators. *ACS Synthetic Biology*, 11(2), pp.780-787.

(3) Yahiro, W., Aubert-Kato, N. and Hagiya, M., 2018, July. A reservoir computing approach for molecular computing. In *Artificial Life Conference Proceedings* (pp. 31-38). One Rogers Street, Cambridge, MA 02142-1209, USA journals-info@ mit. edu: MIT Press.

(4) Nguyen, H., Banda, P., Stefanovic, D. and Teuscher, C., 2020, July. Reservoir computing with random chemical systems. In *Artificial Life Conference Proceedings 32* (pp. 491-499). One Rogers Street, Cambridge, MA 02142-1209, USA journals-info@ mit. edu: MIT Press.

To improve the manuscript, I'd suggest the authors provide a direct comparison with the chemical reservoirs that were previously published, including the same tasks.

We thank the reviewer for pointing out these examples of chemical reservoir computation. We maintain that our work is novel, as it is fundamentally different to the systems described in these reports.

Firstly, if viewed as a reservoir computer, our system is implemented physically, a feat which none of these papers demonstrates. Secondly, our system is based on simple organic compounds, whilst the above-mentioned (theoretical) systems rely upon sophisticated DNA-based chemistry. The principles behind the self-organization of chemical reactions are very different to the underlying base-pair interactions which drive DNA computation. Moreover, although computation using DNA-based systems has been achieved experimentally, reservoir computation specifically has not been demonstrated in these systems, as it remains challenging to synthesise an appropriate DNA-based system. Even implementing a reservoir computer based on the formose reaction via *in silico* simulations is a challenge. The complexity and dynamics of the formose reaction, including processes occurring on widely varying timescales and non-trivial kinetics make it computationally expensive to simulate using modern techniques.

We can also demonstrate that our experimental system can perform with predictive power comparable to the cited, idealised, *in silico* reaction systems. We have implemented a prediction of the second-order dynamical task provided in Liu et al. and originally investigated in Du et al. (*Nature communications*, 2017), using our reaction data and the reported dynamical function and

error metric. The results of this task are shown in Table 1 below, resulting in comparable train and test scores respectively (lower is better) although the *in silico* DNA implementation by Liu et al. scores slightly better. However, it should be noted that from a visual comparison our system does seemingly make predictions closer to the true values than theirs (fig. 3). Unfortunately, we cannot further examine the discrepancy between reported error values and visual performance, as the authors of that study have not made the code for their calculations public. Because we cannot verify the accuracy of their results, we do not include this comparison in our manuscript.

We also do not compare our system to the other above-mentioned in-silico implemented papers as, to the best of our knowledge, these works are not peer-reviewed, nor is their simulation code publicly available.

Table 1: Comparison of Normalized Mean Squared Errors (NMSE) achieved in our work, a memristor array implementation (Du et al. 2017), and an in-silico DNA reservoir (Liu et al. 2022) (lower is better).

	Train error	Test error
Ours	8.6×10^{-4}	8.5×10^{-4}
Du et al. 2017	Not reported	3.6×10^{-3}
Liu et al. 2022	2.4×10^{-4}	3.1×10^{-4}

Fig. 3 a) Reprinted (adapted) with permission from Liu, X. and Parhi, K.K. Reservoir Computing Using DNA Oscillators. ACS Synthetic Biology, 11(2), 780-787 (2022). Copyright 2022 American Chemical Society (figure 7): Solving a second-order nonlinear dynamical task. (left) Theoretical output (blue solid line) and experimentally reconstructed output from the DNA oscillator reservoir computing system (orange dots), for the last 50 time steps from the random training set. (right) Theoretical output (blue solid line) and experimentally predicted output from the DNA oscillator reservoir computing system (red dots), for the last 50 time steps from the random test set. B) Ours, from this work. Training and prediction of the second-order dynamical task using a fluctuating input of the same order and summary statistics as reported in (Du 2017; Liu 2022) . (left) Plot of true and reconstructed signal for a training set of 50 time steps. The first 20 minutes of measurements are not used, because of reactor equilibration. (right) Plot of true and

reconstructed signal for a test set for 50 time steps. The blue line shows the true (target) output, the orange line the prediction after training.

Several times the authors point to the fact that their implementation demonstrates that biological and chemical systems can perform a variety of computational tasks. I don't think this should come as a surprise to anyone. I also don't think it's surprising that the formose reaction can be used to perform complex information processing because almost any physical system can do so when used as a reservoir. The bar for "complex information processing" (whatever that actually means) is really pretty low when it comes to reservoir computing.

Although it has been hypothesised, *experimental* verification of information processing in chemical networks has to our knowledge not been shown before and is, as discussed above, far from self-evident. To demonstrate information processing, it was necessary for us to select an appropriate chemical reaction system (the formose reaction), conditions under which it may actually process the desired information inputs in a relevant manner, and finally an analytical method which gives the time and compound resolution required to observe its information processing. These components of our work are certainly not trivial and could not have been predicted by any model.

Quantifying how information propagation through the system was a neat addition, not usually provided by the physical reservoir computing community. However, a key question that remains unexplored: what are the computational limits of a formose reaction reservoir? How complex of a task can one solve?

To the best of our knowledge, no good objective computation metric exists to determine the computational limits of physical implementations of reservoir computers. All such metrics require access to an *in silico* model of the system to explore parameter regimes and/or perform many repeat runs. As the formose system itself is not computationally tractable (e.g. it cannot be simulated), computing an absolute upper limit is not possible; new, unexplored experimental conditions may result in previously unobserved behaviour, and the direct reliance on an experimental setup means only a finite number of runs can be performed in a reasonable amount of time. The main computational limitations of our system currently originate from the practical setup, not the potential theoretical limitations of the underlying chemistry. Some current limitations are caused by the limited memory present in the reservoir. This is for example shown in the *E. coli* metabolism prediction task, where the behaviour of some specific compounds is not predicted correctly, due to accumulation or degradation reactions over longer timescales than are present in the formose reservoir. We mention this limitation in our manuscript (page 11). Additional limitations arise from the relatively low number of inputs currently used in the system (only 4 may be used simultaneously), due to compatibility issues between the number of syringes, microfluidics setup and reactor.

And how would you scale up such a system to solve more complex tasks? The authors mention the potential for large-scale computation, but it is unclear how one would get there with their approach. Perhaps the authors could try to deduce an upper limit by using their mutual information approach?

In our conclusion we explain possible ways to scale up the system for more complex tasks and larger memory effects. This includes: the inclusion of different initiators (which increases the potential input space); the introduction of different types of chemistries such as phosphorylated compounds, cyanide-based chemistry, or minerals (which all potentially further increase the network size/internal state space, and introduce dynamics at longer timescales). Additionally, compartmentalization using droplet-based reactors may further boost memory-effects. So far, these types of chemistry have mainly been explored in the context of origins-of-life research or systems chemistry, but we believe they may be fruitful starting points for further advancements in chemical reservoir computation. We apologize for not properly including references to relevant research in this direction. To further clarify how scale-up can be achieved, we have now included appropriate references in paragraph 2 of the conclusion accordingly.

>>> Added/modified text in yellow (p15): "In the future, the inclusion of different initiators, such as glycolaldehyde or erythrose, can increase the number of available inputs into the system[6], and different chemistries like phosphorylated and cyanide-based compounds may further diversify complex reaction outputs [44,45]. Work is already underway to establish a direct feedback mechanism between detector and flow controller, enabling autonomously functioning systems capable of unsupervised learning and prediction. Finally, the incorporation of chemistries involving longer timescales, such as the inclusion of mineral[46] or even different phases may be used to extend the memory of the formose reaction [47], enabling longer time series forecasting and other computational tasks requiring long-term memory."

Where we added the following references:

44. Wołos, A. et al. Synthetic connectivity, emergence, and self-regeneration in the network of prebiotic chemistry. *Science* 369, (2020).
45. Ritson, D. & Sutherland, J. D. Prebiotic synthesis of simple sugars by photoredox systems chemistry. *Nature Chemistry* 4, 895–899 (2012).
46. Colón-Santos, S., Cooper, G. J. T. & Cronin, L. Taming the Combinatorial Explosion of the Formose Reaction via Recursion within Mineral Environments. *ChemSystemsChem* 1, e1900014 (2019).
47. Lu, H. et al. Small-molecule autocatalysis drives compartment growth, competition and reproduction. *Nature Chemistry* 16, 70–78 (2024).

As far as I can tell, the methodology is solid and appropriate methods and data is used. I also found the paper to be very well written. The illustrations are clear and support the paper's narrative.

We thank the reviewer for their kind comments, and hope to have sufficiently clarified the significance and possible impact of our work.

Referee #2 (Remarks to the Author):

A.

The manuscript „Chemical reservoir computation in a self-organizing reaction network” by Mathieu G. Baltussen, Thijs J. de Jong, Quentin Duez, William E. Robinson, and Wilhelm T.S. Huck describes experimental results demonstrating computing with formose reaction. The authors consider 5 reagents that can be used for data input and measure up to 106 ion concentrations to obtain the time evolution of the reservoir. After the measurements, the concentrations are processed by a single artificial neuron to produce the output. To demonstrate the computing potential of the considered medium, the authors consider such problems as the classification of points belonging to a specific 2-dimensional structure, simulations of a dynamical system, and the prediction of time evolution based on previously collected knowledge.

B.

In my opinion, the manuscript can be interesting for a broad spectrum of readers, and it is appropriate for publication in Nature because it shows the experimental realization of a chemical computer that can perform non-trivial operations.

C & F. & H.

I have some remarks and questions that should be resolved before the manuscript is accepted.

Through the manuscript, the authors call the network self-organizing. Could they explain what this self-organization is? In my opinion, the reactions are just determined by the presence of specific reagents in the system.

We thank the reviewer for their time in reviewing our manuscript and for their helpful comments.

The network is self-organizing in the sense that we do not design the individual reaction steps in the network. Instead, depending on the reaction conditions, different network topologies emerge, which in chemical systems is commonly referred to as ‘self-organizing’. We refer to our earlier work, Robinson et al. Nature Chem. 2022 and Van Duppen et al. JACS. 2023, for a detailed analysis of the self-organizing properties of the formose reaction. In contrast to self-organised systems, taking DNA-based, genetic, or metabolic reaction networks as an example, each individual step in these networks is carefully design towards a highly specialised task.

All classification problems used the same set of 132 randomly sampled concentrations of formaldehyde and NaOH. I presume not all points of the test set were classified correctly because always the Φ -coefficient is <1 .

- Could the authors mark correctly and incorrectly classified points in Fig.2 (for example, using different colors as in Entropy, 2022, 24(8), 1054)?

The reviewer is correct in their assumption not all points are classified correctly. In fig. 2 we showed only the classification of the full data used as a training set, with the phi-coefficient calculated as an average over test set scores over different train-test splits (split 127:5, inputs are

repeated 20 times in the test split). We believed that choosing one specific train-test split for visualization would result in a biased figure, as some train-test splits perform perfect classification while other train-test splits are biased towards wrong classification. This is especially the case for points close to the classification boundary. However, as the reviewer made clear with their question, this representation can misrepresent the accuracy of the classifier. We have modified figure 2d in the main text to now show for every point individually the average accuracy of its 20 test-set predictions (also shown below in figure 4). As expected, points near (or right on top of) classification boundaries are more likely to be misclassified when not included in the training set. We have also updated the bar-plot comparisons to show the Φ -accuracy ($= (1+\Phi)/2$) instead of the Φ -score.

Fig. 4 (figure 2d in manuscript + updated caption): Results of reservoir classifications for various classification tasks. Dot locations indicate the corresponding input from fig. 2a, and the colour of every point indicates the average classification accuracy of 20 different leave-5-out test-set predictions of the reservoir after training. Shaded areas indicate the true behaviour of the classification function (red/blue for binary classification, red/green/yellow for tertiary classification). The bar charts below every classification plot show the leave-5-out cross-validated Φ accuracy for the formose reservoir (FRC), the training layer without reservoir (LSVC), and various other machine learning classifiers (SCV: Support Vector Classifier, GP: Gaussian Process classifier, MLP: Multi-layer perceptron, ESN: echo state network), where +1 corresponds to perfect predictions, and 0 to total failure. The dashed line indicates the score achieved by the formose reservoir.

- Have the input data different from those included in the training dataset been used for independent verification of the classifier accuracy?

We apologize for the confusing description of our testing procedure. We have only included stratified train-test splits in the analysis, as the algorithm relies only on a linear support vector classifier, without further tuning any hyperparameters. The cross-validation we use in the text refers to different iterations of train-tests splits, where the Φ -score is calculated separately on the test set for every permutation of the split, and then averaged to obtain the values reported in the manuscript. For every train-test split, the training is done completely independent of the test set.

- It would help if the authors give numbers of both Φ -coefficient and the accuracy ($= (1+\Phi)/2$) for all FRC classifiers.

We have included a table in the supporting information, reporting both the Φ -coefficient and accuracy for every classification task and classifier.(SI table 2)

- What were the radii of circles in Circle, Concentric, and Dots problems?

The radii are respectively ~ 0.35 , (0.4, 0.2), and (~ 0.22) on normalized input ranges of (0, 1). We have added this information to the supporting information (SI section 3.3). Also, the code for generating the classification problems is available in a Jupyter notebook in the GitHub repository linked to under the Code Availability section

- Could the authors explain the significant difference between the accuracy of AND and OR classifiers? The problems are isomorphic, so it is probably related to a small training sample.

The difference between accuracy between both problems is due to the classification problem being first translated to a chemical input space to make it suitable for *in chemico* computation. While the original AND and OR problem spaces are isomorphic, the resulting chemical input space ('problem space') is not, as the axes represent different chemical compounds. We can show this more clearly in the following analysis. By performing a Lasso regression (instead of a ridge regression) we obtain a sparse representation of the weights (shown below in figure 5), highlighting the compounds with the most significant contribution to a specific classification task. Here it is clearly shown that the AND and OR problems achieve classification by contributions of different compounds. This can be understood from a chemical point of view. The lower left quadrant of the input landscape corresponds to low concentrations of both sodium hydroxide and formaldehyde. We know from this work, and previous work (Robinson et al., Environmental conditions drive self-organisation of reaction pathways in complex prebiotic reaction networks. Nature Chemistry 14, 623–631 (2022)), that relatively few compounds are produced under these conditions, leading to lower signal to noise and lower dimensionality in the reservoir response. In turn, these factors contribute to slightly lowering the accuracy of the classification of the OR problem compared to the AND problem.

Fig 5 (adapted from figure S12 in SI): Weights obtained from the Lasso regression training of an AND gate and OR gate task, on a normalized data set. Weights are represented by a vertical bar per compound, with compounds numbered according to table 1 (Supporting Information).

As an example of modeling complex dynamics, the authors demonstrate that the formose reaction can replicate the time evolution calculated using a complex reaction model. The authors use fluctuating DHA as the input because it is the common reagent in both systems. On the basis of these results, they propose the mapping between the measured concentrations in the formose reaction and the calculated concentrations in the E.coli model. A nice match has been observed, so the authors claim the chemical reservoir can compute the model.

But how useful is such a simulator? The authors say,,, by continuing the fluctuating input pattern after the training period, the learned linear mapping allows to use the formose reservoir as an emulator of the dynamic system „. What does the fluctuating $u(t)$ mean for times > 30 min? Do the fluctuations repeat the previous pattern?

We apologize for the lack of clarity in our explanation of the fluctuating input experiments. For times > 30 min. we continue the fluctuating pattern such that the input is still random (non-repeating) but holds the same statistical features (mean and standard deviation of fluctuations) as times < 30 min. Figures on extended fluctuation times are provided in the supporting information (S13-S15) and we have update the manuscript to clearly refer to them (page 11). For convenience, we have also included figures on the extended fluctuations and resulting predictions below (figure 7 and 8).

>>> Added/modified text in yellow (p11): “In figure 3c, time trace comparisons between true and predicted concentrations are shown for respectively pyruvate (Pyr), 3-phosphoglyceric acid (3PG), and the co-factor adenosine monophosphate (AMP), showing how the formose reservoir can closely predict the behaviour of the network, for a training time of 30 minutes and a prediction time of 60 minutes. Extended prediction times up to 90 minutes are shown in figures S21-S23.”

Figure 7 (Fig. S11 in SI): a) Fluctuating input flow profiles used in the prediction of metabolic network behaviour. CaCl₂, formaldehyde, and NaOH were kept constant at a flow rate of 36.25 μ L/min, while DHA was varied. b) Ion signals observed in response to the changing flow inputs. Colours indicate different ion signals.

Species															
ACCOA	G6P	OAA	SUC	icd	HCO3	ACEx	LGLU								
ACO	GAP	PEP	SUCCOA	icdP	P	GLCx	BA								
ACP	GL6P	PGA2	X5P	tal	MG	Px	MCOAp								
AKG	GLX	PGA3	ei	talC3	ASP	ACEx_0	MCOAx								
BPG	ICIT	PGN	eiP	tkl	CYS	Hout	BAp								
CIT	KDPG	PYR	eiia	tklC2	MN	GLCp	BAX								
DAP	MAL	Q	eiiaP	ADP	Hin	Pp	GLYp								
E4P	NAD	QH2	eiicb	AMP	H2O	ACEp	GLYx								
F6P	NADH	R5P	eiicbP	ATP	O2	G3P	DAH								
FDP	NADP	RU5P	hpr	CAMP	FAD	GLY	ACCOA_in								
FUM	NADPH	S7P	hprP	COA	FADH2	MCOA									

Figure 8 (Fig. S12 in SI): Prediction results for all substrates in the metabolic network. True (simulated from *E. coli* model) time series are shown as solid lines, predictions from the formose reservoir as dashed lines (train set between 20-50 minutes, test set between 50 and 180 minutes).

- Does the agreement hold if one considers another $u(t)$ with the same average and dispersion?

The agreement indeed holds, as can be seen in S13-S15 (and figure 8 above) when $u(t)$ is continued with a non-repeating pattern. We have added an explicit reference to these figures in the main manuscript.

>>> Added/modified text in yellow (p11): **“In figure 3c, time trace comparisons between true and predicted concentrations are shown for for respectively pyruvate (Pyr), 3-phosphoglyceric acid (3PG), and the co-factor adenosine monophosphate (AMP), showing how the formose reservoir can closely predict the behaviour of the network, for a training time of 30 minutes and a prediction time of 60 minutes. Extended prediction times up to 90 minutes are shown in figures S21-S23.”**

- Is there any use of the simulator trained with fluctuating $u(t)$ when the input is constant and close to the average?

Constant input will result in a steady-state output, which can in principle be perfectly matched by the reservoir computer. However, in this case the reservoir would not perform any relevant time-dependent information processing.

In my opinion, the part on forecasting applications of the reservoir computer is not clearly presented. According to previously discussed applications, $u(t)$ represents the reagent inflow, and it is a free parameter that can be modified at any time according to the experimentalist's wish. Therefore, there is no argument to expect that the relationship in the form $u(t+dt) = W x(t)$ is valid in general. The situation changes if the function $u(t)$ is fixed before the experiment, and we want to reproduce it from system observations in a time sub-interval shorter than the whole experiment. Our prediction should be accurate if $u(t)$ is periodic and the sub-interval covers the whole period. It leads to complete failure if the selected sub-interval does not cover all representative parts of $u(t)$. In the considered example, the authors assumed that the Lorenz attractor describes changes in $u(t)$, and they focused their attention on restoring this input.

In my opinion, the introduction to forecasting applications that includes the information given above would help the readers to get the idea of forecasting (of course, if I have got the idea right).

We apologize for the lack of clarity in our explanation of the forecasting. The reviewer is correct in their assumption the input $u(t)$ is in principle a free parameter. Indeed, the application of forecasting is only possible when $u(t)$ has temporal structure that allows the relationship $u(t+dt) = W x(t)$ to hold. We have updated the text accordingly.

>>> Added/modified text in yellow (p13): **“For environmental dynamics with a temporal structure (e.g. deterministic dynamics and/or periodicity), we can attempt to forecast changes by learning a linear mapping between the reservoir state $x(t)$ and a future input $u(t + \delta t)$ representing the environment dynamics, as ($u(t + \delta t) = Wx(t)$), during a short training interval as shown in figure**

4a.”

G. OK

We thank the reviewer for their useful feedback and kind comments. We hope to have sufficiently answered the remaining questions and remarks.

Referee #3 (Remarks to the Author):

The manuscript experimentally demonstrates physical reservoir computation using chemical reaction networks related to formose reactions (which are found in synthesizing sugar from formaldehyde). The capability of the chemical reaction network as a physical reservoir is demonstrated for three machine learning information processing tasks, including (i) nonlinear classifications of linearly inseparable data in two classes, (ii) dynamical system approximations for metabolic models, and (iii) autonomous dynamical system emulations with the Lorenz attractor exhibiting chaotic behavior.

This work is directly along the concept of physical reservoir computing which enable machine learning computation by using an untrainable physically-implemented dynamic reservoir and a trainable readout (Tanaka et al., Recent advances in physical reservoir computing: A review, *Neural Networks*, 2019). The idea of using chemical and electrochemical reaction systems as a dynamic reservoir is not so novel as there exist some previous studies (e.g. Kan et al., Physical Implementation of Reservoir Computing through Electrochemical Reaction, *Advanced Sciences*, 2021; Nguyen et al., Reservoir Computing with Random Chemical Systems, *Artificial Life Conference Proceedings*, 2020). It is not surprising to imagine that complex chemical reaction networks as used in the current work may be available as a dynamic reservoir after seeing the previous studies on In-Materio reservoir computing (Dale et al., Reservoir Computing as a Model for In-Materio Computing, *Advances in Unconventional Computing*, 2016). In the reservoir computing field, the impact of the current work is not so high under the existence of a plenty of implementation demonstrations of other types of physical reservoirs (e.g. Torrejon, J. et al. Neuromorphic computing with nanoscale spintronic oscillators. *Nature* 547, 428–431, 2017).

Nevertheless, as far as I know, this work is the first attempt to use the formose reaction network as a dynamic reservoir, presenting the high potential of chemical reservoir computing based on exhaustive experiments. The authors' sophisticated experimental system operating the self-organized chemical reactions (probably developed in their previous study [6]) seems to be the strong point of this work, enabling the inputs of various temporal signals, the control of chemical reaction processes, and the recording of time-varying chemical compound concentrations.

We thank the reviewer for their time in reviewing our manuscript.

Considering that the concept of reservoir computing, the explanation in the schematic overview in Figure 1 is misleading. Normally a dynamic reservoir receives (time-)sequential input signals and reacts to a history of the input signals (not only to the current input signal), and therefore a reservoir computing system is suited for approximating a dynamical system. It is not mathematically correct to state that the target (Figure 1e) is a "function" of the input variable (Figure 1b). Correctly, the target output sequence is a transformation of the input sequence by a "filter" or a "dynamic system". This is clearly described in the seminal papers of reservoir computing (Jaeger, The "echo state" approach to analysing and training recurrent neural networks-with an erratum note', GMD Technical Report, 2001; Maass et al., Real-time computing without stable states: A new framework for neural computation based on perturbations, *Neural Computation*, 2002). A correction of the corresponding description is necessary for explaining why the dynamical system emulation and forecasting (i.e. 2nd

and 3rd tasks) are successful with the chemical “dynamic” reservoir. It should be noted that the classification of the static data (i.e. 1st task) shown in Figure 2 is a special case where the correspondence between an initial state and an equilibrium state of the reservoir is used as a static “function”.

We apologize for not clearly specifying the difference between static functions and dynamic filters in figure 1 and the corresponding text. In our manuscript we tried to strike a balance in understandability and correctness, especially considering the broad range of backgrounds our readership may have (so far, we have found that most chemically inclined audiences benefited from an explanation starting with static function approximations before introducing dynamic tasks, and reflected this in our approach to fig. 1). We did explain the different types of computational tasks considered in the Supporting Information (section 2.4 and 2.5). However, we realize this was not properly reflected in the manuscript and have updated the figure and text to reflect the time-dependent nature of the tasks. We have included those changes below for your convenience.

Updated figure (updated parts in yellow):

>>> Added/modified text in yellow (caption Fig. 1, p5): “Figure 1 – A schematic overview of the formose reservoir computer. a) The formose reaction and its information processing abilities. left: A schematic view of the formose reaction network. Arrows indicate chemical transformations between compounds in the network. Dihydroxyacetone and formaldehyde are used as initial reactants and indicated with purple arrows. right: Graphical summary of the information processing tasks of which the formose reaction is capable. b-f) A schematic overview of the experimental setup and reservoir training process. b) A set of input variables u used to obtain a target **(dynamic) transformation $f(t, u)$** . These input variables are also used as flow inputs into the reservoir. c) Syringe pumps containing the formose reagents (formaldehyde, dihydroxyacetone, sodium hydroxide, and calcium chloride) are connected to the inlets of a CSTR and are used to feed the input into the reservoir. d) The reservoir outlet is connected to an ion mobility-mass spectrometer for online detection of the reservoir composition. The state of the reactor x is measured over time in response to changing inputs. e) The target **(dynamic) transformation $f(t, u)$** obtained from the input. f) Weights W are trained on the states of the reservoir to obtain an approximation to the target function, which can then be used for further predictions.”

>>> Added/modified text in yellow (p5): “Our chemical reservoir computer is built around the formose reaction (fig. 1a) in a continuous stirred tank reactor (CSTR) (fig. S1-S2). Following the reservoir computation paradigm³⁴, we can approximate **any target (dynamic) transformation (f) under the influence of a set of input variables (u)** (fig. 1b and fig. 1e) by feeding the input variables as a sequence of chemical concentrations into the reservoir (fig. 1c). We investigated three kinds of **target transformations**: analytical expressions in the form of classification tasks, integral solutions of differential equations, and chaotic maps in the form of time series forecasts of the Lorenz system. **The first type of task only uses the steady-state features of the reservoir to approximate a static function, while the other types employ the full dynamics of the reservoir to approximate different kinds of dynamic systems.**”

In addition, it does not make sense to compare the proposed method with the ESN (Echo State Network) because the classification performance of the ESN depends on the order of samples given to the reservoir but the order is meaningless in this task. My concern is that the authors do not care about the (mathematical) difference between an approximation of a “static function” and that of a “dynamic filter”.

For the inclusion of the ESN in the classification comparison, we have been careful to use an implementation suitable for static classification tasks (PyRCN ESNClassifier, as found on GitHub https://github.com/TUD-STKS/PyRCN/blob/main/src/pyrcn/echo_state_network/esn.py#L619). To the best of our knowledge, the performance of this classifier does not depend on the order of samples as it only uses the steady-state of the reservoir for classification, and we have indeed observed equal performance independent of input order. Furthermore, the code implementing the classification comparison (and generating the results in figure 2) is available in a Jupyter notebook (analysis/classification.ipynb) in this manuscript’s GitHub repository as linked to under the Code Availability section. We have updated mentions of the ESN in the manuscript to reflect it is used explicitly as an echo state classifier (ESC).

The future perspective of this work is not fully understandable. It is reasonable that the chemical reaction networks are compatible with biological networks. However, the readout computation (i.e. an error minimization with a ridge regression) is currently done with a general-purpose computer. It is not clear how biological systems can implement the regression algorithms.

To further clarify how a chemical readout could be performed using a chemical or biological system, we have added an additional analysis of the nonlinear classification task, where we use a Lasso regression method to obtain a sparse representation of the weights need for computation (see figure 9 below). These weights indicate the chemical compounds most relevant to each specific task and could in principle be selected for during a secondary chemical process which translates these specific compounds to a singular output. We have included here an additional figure showing how such a process approximates an XOR gate if specific to a sufficient amount of compounds (see figure 10 below, extended set in figure 2. Additionally, we have performed proof-of-principle experiments on a grid of inputs using selective reagents (see figure 11 below). While not selective for a single compound, these reagents react with certain sets of compounds generated by the formose reaction to produce a colorimetric response. These visible outputs could then be overlaid to produce a combined readout directly from chemical reactions. We have modified our conclusion to further explain the potential of such a chemical readout (see updated text below).

Fig 9 (adapted from figure S12 in SI): Weights obtained from the Lasso regression training of an XOR gate classification task, on a normalized data set. Weights are represented by a vertical bar per compound, with compounds numbered according to table 1 (Supporting Information).

Fig 10 (S13-17 in SI, shortened version included here): Visualization of the creation of an XOR gate by composition of different compounds with weights as obtained through a lasso regression (see figure 2 above for all the weights), ordered from left-to-right by the absolute value of weights. Every fifth compound is shown here for brevity, a full version of 25 compounds is included in the Supporting Information. a) Row of scatter plots showing measured intensities for 5 different compounds (normalized values, compounds denoted by X_{xx}) and a shaded background obtained by interpolation. Plots are sorted by decreasing absolute weight value (e.g. the contribution in the classification process). Axis are the same as in manuscript figure 2, with NaOH input on the x-axis and formaldehyde input on the y-axis. b) For every compound in a), the measured intensities are multiplied by a weight value (denoted by W_{xx}) obtained from the lasso regression to obtain a weighted outputs for all 5 compounds. c) A set of five plots showing from left to right the sum of the weighted outputs for an increasing number of compounds (respectively 1, 6, 11, 16, and 21 compounds, ordered by absolute value of weight). By including more compounds in the summed output, an approximate XOR classification response is created.

Fig. 11 (Supporting Information figure S19): Schematic overview and results of classification using colorimetric reagents. a) A scatter plot showing the sampled DHA and NaOH concentrations. **b)** each sample consists of a unique composition of compounds, depending on the conditions each compound has a specific reactivity with the reagent, this can also be viewed as a set of weights. **c)** In isolation each compound would produce a different color, depending on its concentration and reactivity with the reagent. These isolated contributions cannot be observed, instead we observe one final output colour that can be considered a sum of the individual effects. **d)** A example final readout. **e)** Result of visual readout using Benedict's reagent after 5 minutes. **f)** Result of colorimetric test using Seliwanoff's resorcinol reagent after 1 hour 42 minutes. **g)** Result of colorimetric test using Seliwanoff's thymol reagent after 1 hour 55 minutes.

Furthermore, we do not necessarily envision a full chemical reservoir computer (consisting of both chemical and *in silico* components) to interface with biological systems. Instead, we expect that the chemical network itself (without the *in silico* readout layer) could function as an effective interface, both to perform sophisticated readout of cellular states and perform communication with the cell. This contrasts with modern biological interfaces - in the form of enzymatic assays, qPCR, RNA-seq, and other Omics techniques - which all essentially form a direct, but sometimes limited, readout layer. Complex, self-organizing chemical networks have been little explored in this context, but the information processing capacity shown in our work demonstrates their potential. It has been previously indicated this is indeed possible in for example quorum sensing pathways in bacteria (Gardner et al. Sugar synthesis in a protocellular model leads to a cell signalling response in bacteria. *Nature Chemistry* 1, 377–383 (2009)). We have updated our conclusion to better clarify this potential application of our work.

>>> Added/modified text in yellow (p15-16): "Importantly, while the reservoir computation paradigm is used in this work to establish and exploit the information processing capabilities of the formose reaction, in the future this electronic 'readout layer' may be replaced, or altogether omitted, by interfaces and reservoirs tailored to specific applications. So far, physical reservoir computers always depend on *in silico* training of the readout layer. A significant next step for in chemico computation would be a fully chemical readout capable of autonomous learning. Let us therefore present a first proof-of-concept experiment, in which we show how simple reagents

that react with components of the reaction mixture, can be added to the output of the formose reservoir to chemically set readout weights. The overall sum of the concentration of compounds in the mixture 'multiplied' by the reaction with the added reagent results in a colorimetric response (see SI sections 1.3 and 3.7, and figure S19). As the figure shows, each combination of input parameters results in a specific hue or colour. Thus, the chemical readout allows for the identification of different environmental inputs, certainly if combinations of different reagents or reaction times are used. Potentially, this approach may be further extended to incorporate a feedback mechanism that can change the amount and type of reagents and modify other system hyperparameters to perform a specific computational task, either via the inclusion of an in-the-loop computer or directly via physical learning[48]. The information processing abilities of the formose reaction, and potentially other self-organizing chemical networks, may offer a powerful interface with biological systems, allowing for direct chemical communication with cellular signalling pathways. In fact, the formose reaction has already been used to communicate with cells, stimulating bioluminescent responses via interaction with a quorum sensing pathway in the marine bacterium *Vibrio harveyi*⁴². The approach presented here could potentially be adapted to control such cellular responses in sophisticated manners by communicating with cells in their own 'language' via a direct chemical interface. Such an interface would allow us to establish a new class of intelligent matter, driven directly by the flux of information through chemical reaction networks.

Where we added the following references:

48. Stern, M. & Murugan, A. Learning Without Neurons in Physical Systems. *Annual Review of Condensed Matter Physics* 14, 417–441 (2023).

As for the practicality of the proposed method, I wonder if it is sufficient to test the chemical reservoir computation only with the artificial data generated by deterministic functions/dynamical systems. The capability of the proposed method for real data processing is still unclear.

We agree with the reviewers comment on the real-world applicability of our method. We specifically chose the task of predicting the behaviour of a metabolic network as a demonstration of the formose reactions predictive capabilities, because the network and parameters we use are based on measurements and best fits to *in vivo* data (Oliveira et al. *Dynamical nonequilibrium molecular dynamics reveals the structural basis for allostery and signal propagation in biomolecular systems. The European Physical Journal B* 94, (2021)). We feel that this task is a good proxy towards tasks more specific to the real world, while retaining the computational tractability necessary for in-depth analysis of the results.

We thank the reviewer for their useful feedback and kind comments. We hope to have sufficiently answered the remaining questions and remarks, and explained the outlook and future perspective of our work.

Reviewer Reports on the First Revision:

Referees' comments:

Referee #1 (Remarks to the Author):

I would like to thank the authors for their effort to address all reviewer concerns. The paper has much improved, but my fundamental concerns re: novelty and intellectual merit remain. Yes, this is a very first chemical RC implementation that is unique in its own way. But so what? Is it not clear to me how such an implementation (of the reservoir) only actually advances the knowledge and science in this field. Is there anything your RC can do that other RCs can't do?

There's been tons of physical RC implementations recently. Some can be called "chemical" of some sorts, e.g.,

Conducting polymers: <https://arxiv.org/pdf/2001.04342.pdf>

Electrochemical: <https://onlinelibrary.wiley.com/doi/10.1002/adv.202104076>

And yes, most implement the output layer outside of the actual physical substrate. I have the exact same concerns for the entire body of work in that area. The key challenge does not consist in showing that a physical substrate can be used for computation, the key challenge is to implement the entire RC, with its output layer, in the substrate.

I respectfully disagree that "the current dependence on in silico training is present in all fields of physical reservoir computation and is therefore a major open challenge on its own." Yes, it's a challenge, but there are now most definitely various examples, e.g., of memristive systems, where learning is done in the "substrate."

I never said your implementation was "trivial" and I apologize if I gave that impression. What I wanted to emphasize is that most physical systems can be used as reservoirs. That is why I don't believe there is much novelty in advancing the field by demonstrating yet another physical implementation of a reservoir. But perhaps that is more of a philosophical discussion.

To the best of my knowledge, Banda's (<http://dx.doi.org/10.1098/rsif.2013.1100>) work is the only chemical RC work that "implements" the output layer in the same substrate. Yes, it's theoretical and not experimental, nevertheless. I commend your addition of the proof of principle/demonstration of a possible implementation, but I really wish you'd have gone all the way to an actual implementation. As it is, the paper can only be added to the growing list of yet another RC implementation.

"Firstly, if viewed as a reservoir computer, our system is implemented physically, a feat which none of these papers demonstrates."

Agreed, but again, how did you advance the science of RCs? While I'm impressed by your implementation, I fail to see the practical value of such a physical implementation. The same applies to other physical implementations (VLSI, FPGA, memristors, spintronics, you name it...). As far as I'm concerned, they rarely advance the science and knowledge.

In summary: it seems we disagree on how much of a breakthrough your work represents. Thankfully we don't have to agree on that aspect. As I was trying to say above, I'm impressed by your work and it's far from trivial, quite the opposite.

Referee #2 (Remarks to the Author):

A.

This is the report on the second version of the manuscript; thus, I do not repeat what was the subject of research. Just in short, the authors consider reservoir computing with the formose reaction. They measure the time evolution of concentrations of over 100 ions and map those concentrations into the computation output. The usefulness of the selected chemical medium is demonstrated by considering such examples as the determination of geometrical objects within a unit square, simulations of a dynamical system, and the prediction of future inflows based on previously collected knowledge. In my opinion, the applicability of the considered medium to the classification problem is not surprising, especially when the number of train cases is similar to the dimension of the recorded signal. However, the fact that the formose reaction can be used to simulate another chemical network or to predict future inflows is surprising and requires more careful analysis.

B.

In my opinion, the manuscript can be interesting for a broad spectrum of readers, and it is appropriate for publication in Nature because it shows the experimental realization of a chemical computer that can perform non-trivial operations. Reading the answers to the other reviewers, I believe the authors are unaware of fully autonomous image processing with the photosensitive variant of BZ-reaction demonstrated by Kuhnert, Agladze, and Krinsky almost 35 years ago (Nature 337, 244–247).

C & F. & H.

In my opinion, the manuscript is still not in publishable form, and the authors should improve both the presentation of results and the quality of the description.

I said above that the fact that formose reaction can be used to simulate another system or predict future inflows is surprising and requires more careful analysis. I think computation should give a quantitative result. However, figures 3c and 4b, proving the usefulness of the method, show a qualitative similarity between the anticipated result and its prediction.

In my opinion, the authors should present the relative difference between the anticipated result and its prediction.

In the case of simulations, it would be:

$$(\text{conc}(t) - \text{sim}(t)) / \text{sim}(t)$$

where $\text{conc}(t)$ is the concentration predicted on the basis of formose reaction and $\text{sim}(t)$ is the result of simulations.

Similarly, in the case of inflow prediction, the relevant quantity is

$$(\text{fore}(t) - \text{infl}(t)) / \text{infl}(t)$$

where $\text{infl}(t)$ is the true inflow to the reservoir and $\text{fore}(t)$ is its forecast.

The presentation of results requires additional information. Many essential facts are in the SI. In my opinion, a few sentences added to the manuscript will make it easier to understand.

Here are more specific questions:

Nonlinear classification:

On p.5, the authors say: "Inputs to the reservoir are controlled by changing flow rates. „ So why are the input units in Fig. 2A refer to concentrations, not flows?"

Does the reactor converge to a stationary stable state for all inputs? The p. 7 sentence: „The final output of the foremost reservoir was obtained as the averaged ion intensities over the final 10-minute sample period of the steady-state output, „ suggests that the steady state was not reached because, if so, no average is needed.

The jargon „leave-5-out" should be explained in the main text, not only in SI.

Does the accuracy of a single „leave-5-out" case include both train and test points?

Complex dynamics:

Replace input in Fig. 3b with Fig. S20a.

Forecast:

Does W depend directly on time in the relationship $u(t+dt) = W x(t)$?

Was the density of data points the same as that of complex dynamics?

Were all collected data points (how many per single ion) used to obtain W ?

Referee #3 (Remarks to the Author):

The authors carefully and appropriately addressed most of my questions and comments.

In particular, the additional analysis on the readout weights optimized with Lasso regression with L1-norm regularization suggest the potential of a chemical readout. The result is effective for enhancing the significance of this work and valuable toward realizing all-chemical reservoir computing in the future.

For the authors' response to my comment on the comparison with echo state network (rephrased as echo state classifier in the revised manuscript) in Fig.2d, I still have a feeling of strangeness. In my opinion, applying ESNClassifier in PyRCN to static data classification is not appropriate even though it is possible, because in such a case the reservoir state influenced by the history of input sequence is not necessary and could be harmful. In fact, the performance of ESN is relatively low in Fig.2d. The other model found in PyRCN, ELM, is rather suited for this task, considering the mechanism of the model. I wonder if it is meaningful to adopt the ESN implemented in an exceptional way for performance comparison. There is a concern that general readers misunderstand the essence of RC and expert readers familiar with RC are confused by looking this figure.

Referee #3 (Remarks on code availability):

The codes are useful for reproducing many results in this manuscript.

Author Rebuttals to First Revision:

Referee #1 (Remarks to the Author):

I would like to thank the authors for their effort to address all reviewer concerns. The paper has much improved, but my fundamental concerns re: novelty and intellectual merit remain. Yes, this is a very first chemical RC implementation that is unique in its own way. But so what? Is it not clear to me how such an implementation (of the reservoir) only actually advances the knowledge and science in this field. Is there anything your RC can do that other RCs can't do?

We believe the impact of the work is in the field of chemistry. There has been a long-standing interest in developing chemical systems that have molecular information processing capabilities. These attempts have not yet resulted in chemical systems with capabilities that are comparable to electronic/in silico systems. Our work is the first demonstration that molecular information processing systems can be constructed using the RC paradigm, and their performance is on par with in silico counterparts.

There's been tons of physical RC implementations recently. Some can be called "chemical" of some sorts, e.g.,

Conducting polymers: <https://arxiv.org/pdf/2001.04342.pdf>

Electrochemical: <https://onlinelibrary.wiley.com/doi/10.1002/adv.202104076>

And yes, most implement the output layer outside of the actual physical substrate. I have the exact same concerns for the entire body of work in that area. The key challenge does not consist in showing that a physical substrate can be used for computation, the key challenge is to implement the entire RC, with its output layer, in the substrate.

We have supplied data in the manuscript ([colorimetric readout]) which clearly demonstrate how a direct chemical readout can be obtained from the formose reaction. Furthermore, we have also demonstrated the output for certain tasks can be directly read out from the formose reaction by observing a small subset of compounds ([lasso regression figure/discussion]).

I respectfully disagree that "the current dependence on in silico training is present in all fields of physical reservoir computation and is therefore a major open challenge on its own." Yes, it's a challenge, but there are now most definitely various examples, e.g., of memristive systems, where learning is done in the "substrate."

I never said your implementation was "trivial" and I apologize if I gave that impression. What I wanted to emphasize is that most physical systems can be used as reservoirs. That is why I don't believe there is much novelty in advancing the field by demonstrating yet another physical implementation of a reservoir. But perhaps that is more of a philosophical discussion.

To the best of my knowledge, Banda's (<http://dx.doi.org/10.1098/rsif.2013.1100>) work is the only chemical RC work that "implements" the output layer in the same substrate. Yes, it's theoretical and not experimental, nevertheless. I commend your addition of the proof of principle/demonstration of a possible implementation, but I really wish you'd have gone all the way to an actual implementation. As it is, the paper can only be added to the growing list of yet another RC implementation.

"Firstly, if viewed as a reservoir computer, our system is implemented physically, a feat which none of these papers demonstrates."

Agreed, but again, how did you advance the science of RCs? While I'm impressed by your implementation, I fail to see the practical value of such a physical implementation. The same applies to other physical implementations (VLSI, FPGA, memristors, spintronics, you name it...). As far as I'm concerned, they rarely advance the science and knowledge.

In summary: it seems we disagree on how much of a breakthrough your work represents. Thankfully we don't have to agree on that aspect. As I was trying to say above, I'm impressed by your work and it's far from trivial, quite the opposite.

We thank the reviewer for their time and effort in reviewing our manuscript and appreciate their respectful discourse.

Referee #2 (Remarks to the Author):

A.

This is the report on the second version of the manuscript; thus, I do not repeat what was the subject of research. Just in short, the authors consider reservoir computing with the formose reaction. They measure the time evolution of concentrations of over 100 ions and map those concentrations into the computation output. The usefulness of the selected chemical medium is demonstrated by considering such examples as the determination of geometrical objects within a unit square, simulations of a dynamical system, and the prediction of future inflows based on previously collected knowledge. In my opinion, the applicability of the considered medium to the classification problem is not surprising, especially when the number of train cases is similar to the dimension of the recorded signal. However, the fact that the formose reaction can be used to simulate another chemical network or to predict future inflows is surprising and requires more careful analysis.

We thank the reviewer for their time in assessing the second version of our manuscript. We agree with the reviewer that the first application we show, on the classification problems, is not necessarily surprising. However, we have been positively surprised by the performance of our reservoir in this regard in comparison to other classification methods, especially considering that we have not performed any finetuning of experimental parameters for these tasks specifically. Many previously explored chemical systems also tend towards a relatively low-dimensional effective state space, which would make them less suitable for similar tasks. As for the dynamic tasks, we have provided additional analysis to your comments below, and we intend to investigate the potential of these in much more depth in future work, in the formose reaction and other complex chemical networks.

B.

In my opinion, the manuscript can be interesting for a broad spectrum of readers, and it is appropriate for publication in Nature because it shows the experimental realization of a chemical computer that can perform non-trivial operations. Reading the answers to the other reviewers, I believe the authors are unaware of fully autonomous image processing with the photosensitive variant of BZ-reaction demonstrated by Kuhnert, Agladze, and Krinsky almost 35 years ago (Nature 337, 244–247).

We thank the reviewer for pointing out this early example of chemical image processing, of which we were indeed unaware.

C & F. & H.

In my opinion, the manuscript is still not in publishable form, and the authors should improve both the presentation of results and the quality of the description.

I said above that the fact that formose reaction can be used to simulate another system or predict future inflows is surprising and requires more careful analysis. I think computation should give a quantitative result. However, figures 3c and 4b, proving the usefulness of the method, show a qualitative similarity between the anticipated result and its prediction.

In my opinion, the authors should present the relative difference between the anticipated result and its prediction.

In the case of simulations, it would be:

$$(\text{conc}(t) - \text{sim}(t)) / \text{sim}(t)$$

where $\text{conc}(t)$ is the concentration predicted on the basis of formose reaction and $\text{sim}(t)$ is the result of simulations.

Similarly, in the case of inflow prediction, the relevant quantity is

$$(\text{fore}(t) - \text{infl}(t)) / \text{infl}(t)$$

where $\text{infl}(t)$ is the true inflow to the reservoir and $\text{fore}(t)$ is its forecast.

For both cases we have added quantitative error measures to the figures in the main manuscript (reproduced in figure 1 and 2 below). We have opted for a slightly different error metric than the reviewer suggested, as the various scales involved in our data would lead to very different error scales. To incorporate scale invariance, we use the Absolute Scaled Error instead:

$$ASE = \frac{|y_{true}(t) - y_{pred}(t)|}{\frac{1}{T_{train}} \sum_t |y_{true}(t) - \bar{y}_{true}|}$$

With \bar{y}_{true} the mean of the training data. This metric better accommodates the various scales on which predictions are made. It can be interpreted as a comparison between the reservoir predictor and a naive mean-data predictor (replacing all predictions with the mean of the training data). A score below 1 implies the reservoir predictor performs better. We think this is a slightly more appropriate error metric, because it can be used to directly compare different predictions across scales. We have also added an explanation of this error metric to the Methods section.

Figure 1 (adaptated from fig. 3 main text): Three example predictions for the behaviour of pyruvate, 3-phosphoglyceric acid and adenosine monophosphate (AMP). The red lines indicate the ‘true’ simulated response of the model. The yellow lines show part of the train set, and the blue lines the predictions of the formose reservoir after training. Indicated times on the x-axes are both the physical reservoir time and the model time. The Absolute Scaled Error (ASE, Methods) over time is shown below the predictions.

Figure 2 (adapted from fig. 4 main text): Time traces, error plots, and comparison plots for forecasts of simultaneously varying DHA, NaOH and formaldehyde inputs that resemble the behaviour of a Lorenz attractor. True inputs are shown as purple, orange and red lines respectively, and the forecasts ($\delta t = 120$ s) as blue lines. The Absolute Scaled Errors (ASE, Methods) over time are shown below the predictions.

The presentation of results requires additional information. Many essential facts are in the SI. In my opinion, a few sentences added to the manuscript will make it easier to understand.

Here are more specific questions:

Nonlinear classification:

On p.5, the authors say:” Inputs to the reservoir are controlled by changing flow rates. „ So why are the input units in Fig. 2A refer to concentrations, not flows?

We apologize for the confusing description of this part of the experimental setup. For the flow reactor, we change the concentration of input species by changing the flowrates, so input concentrations and flowrates refer to the same quantities, only described in different units. We have gone through the manuscript to further clarify this where necessary. We've also added additional explanation to the Methods section

Changed text: p3 *"Input concentrations to the reservoir are controlled by changing flow rates"*

Added Methods: *"Inputs to the reactor were controlled by changing flow rates of selected syringes. For a desired input concentration C_{in} , the flowrate can be calculated as $F = F_{tot} C_{in} / C_{syr}$, with F_{tot} the total flowrate of the system (217.5 $\mu\text{l}/\text{min}$ in all experiments, corresponding to a residence time of 2 minutes) and C_{syr} the concentration of the selected syringe."*

Does the reactor converge to a stationary stable state for all inputs? The p. 7 sentence: „The final output of the foremost reservoir was obtained as the averaged ion intensities over the final 10-minute sample period of the steady-state output, „ suggests that the steady state was not reached because, if so, no average is needed.

We do indeed reach a steady-state for all inputs. We average over the final 10 minutes of the sampling period because mass-spectrometer measurements can be relatively noisy (especially for high-intensity signals).

The jargon „leave-5-out“ should be explained in the main text, not only in SI.

We have modified the description of our validation procedure to make it clearer. Due to length constrained for the main text, we have also added additional explanation directly in the Methods section:

Main text (p4, changes in yellow)

"This training procedure was performed for a variety of nonlinear classification tasks (fig. 2d) and validated by calculating the average Φ -accuracy for 520 different leave-5-out train-test splits (Methods). The reported Φ -accuracies are the averages over all test-sets and can be found in Extended Data Table 1."

Methods (p17, changes in yellow):

"For every task, a stratified leave-5-out cross validation was performed with 520 repeats in total, with every input as part of the test set 20 times (20 repeats of 26 random splits, 5 inputs per split), and the Φ score was calculated over the test set for every repeat as

$$\Phi = (TP \times TN - FP \times FN) / \sqrt{(TP + FP)(TP + FN)(TN + FP)(TN + FN)}$$

where TP denotes the number of true positives, TN denotes true negatives, FP denotes false positives, and FN denotes false negatives. This score returns +1 for perfect predictions, and -1 for completely wrong predictions. The reported Φ -accuracy was then obtained as $(\Phi + 1)/2$, and averaged over all 520 repeats. More information is available in SI section 3.2-3.6 and code is provided in the analysis/classification.ipynb notebook on Github.

Does the accuracy of a single „leave-5-out” case include both train and test points?

Accuracies are calculated only from the test-points, not the train-set. We have clarified this in the manuscript, as shown in the text above.

Complex dynamics:

Replace input in Fig. 3b with Fig. S20a.

We have replaced the figure of the input, and included additional labelling/units on the axes.

Forecast:

Does W depend directly on time in the relationship $u(t+dt) = W x(t)$?

W remains independent of time for the full train and test sets. We have clarified this in the manuscript.

Updated text (p7): *“For environmental dynamics with a temporal structure (e.g. deterministic dynamics and/or periodicity), we can attempt to forecast changes by learning a linear mapping between the reservoir state $x(t)$ and a future input $u(t + \delta t)$ representing the environment dynamics, as $u(t + \delta t) = Wx(t)$, during a short training interval as shown in figure 4a. Here, W denotes the static weights obtained during the training phase.”*

Was the density of data points the same as that of complex dynamics?

During all experiments, data points were collected every 500ms. To reduce noise in the final dataset, output was averaged in bins of 10 seconds. This is further explained in the Methods section.

Were all collected data points (how many per single ion) used to obtain W ?

W was obtained by training on 20 minutes of collected data points (2400 data points per ion, 120 points per ion after binning). All ions were used in this training.

Referee #3 (Remarks to the Author):

The authors carefully and appropriately addressed most of my questions and comments.

We thank the reviewer for their time in assessing the second version of our manuscript.

In particular, the additional analysis on the readout weights optimized with Lasso regression with L1-norm regularization suggest the potential of a chemical readout. The result is effective for enhancing the significance of this work and valuable toward realizing all-chemical reservoir computing in the future.

For the authors' response to my comment on the comparison with echo state network (rephrased as echo state classifier in the revised manuscript) in Fig.2d, I still have a feeling of strangeness. In my opinion, applying ESNClassifier in PyRCN to static data classification is not appropriate even though it is possible, because in such a case the reservoir state influenced by the history of input sequence is not necessary and could be harmful. In fact, the performance of ESN is relatively low in Fig.2d. The other model found in PyRCN, ELM, is rather suited for this task, considering the mechanism of the model. I wonder if it is meaningful to adopt the ESN implemented in an exceptional way for performance comparison. There is a concern that general readers misunderstand the essence of RC and expert readers familiar with RC are confused by looking this figure.

We appreciate the reviewer's concern for the comparison of classification algorithms. We agree that a comparison to ESNs may be confusing and/or incorrect at this point in the manuscript. We do think this specific implementation of the ESN classifier is suitable for this type of task and have confirmed input order does not change results. However, as we cannot completely guarantee this reservoir functions correctly (or optimally) in this setting, following the reviewer suggestions, we have replaced it with the ELM from the same Python package. As expected, this implementation does perform better than the ESN, and is comparable to the formose reservoir, SVC and MLP in accuracy. We have adapted the figure and text correspondingly (see also the figure below). These changes do not impact on our overall conclusions in the manuscript.

As a side note, from a conceptual perspective the formose reservoir is of course more similar to the ESN implementation, due to its dynamic and recurrent nature, in comparison to the strictly feed-forward nature of the ELM. This similarity inspired our original choice for the ESN comparison.

Figure 3 (Adapted from fig. 2 main text). The last bar of the accuracy-plots has been replaced with an Extreme Learning Machine (ELM) classifier.

Referee #3 (Remarks on code availability):

The codes are useful for reproducing many results in this manuscript.